# FROM DATA STATISTICS TO FEATURE GEOMETRY: HOW CORRELATIONS SHAPE SUPERPOSITION

**Lucas Prieto , Edward Stevinson , Melih Barsbey , Tolga Birdal** [*]**, Pedro A.M. Mediano**[*]
Imperial College London

## ABSTRACT

A central idea in mechanistic interpretability is that neural networks represent more features than they have dimensions, arranging them in superposition to form an over-complete basis. This framing has been influential, motivating dictionary learning approaches such as sparse autoencoders. However, superposition has mostly been studied in idealized settings where features are sparse and uncorrelated. In these settings, superposition is typically understood as introducing interference that must be minimized geometrically and filtered out by non-linearities such as ReLUs, yielding local structures like regular polytopes. We show that this account is incomplete for realistic data by introducing Bag-of-Words Superposition (BOWS), a controlled setting to encode binary bag-of-words representations of internet text in superposition. Using BOWS, we find that when features are correlated, interference can be constructive rather than just noise to be filtered out. This is achieved by arranging features according to their co-activation patterns, making interference between active features constructive, while still using ReLUs to avoid false positives. We show that this kind of arrangement is more prevalent in models trained with weight decay and naturally gives rise to semantic clusters and cyclical structures which have been observed in real language models yet were not explained by the standard picture of superposition. Code for this paper can be found at: `https://github.com/LucasPrietoAl/correlations-feature-geometry`.

## 1 INTRODUCTION

The field of mechanistic interpretability (MI) aims to understand deep learning models by decomposing them into interpretable components such as features or circuits, and understanding how these interact (Olah et al., 2020). A central idea in this field is that models represent more features than they have dimensions, arranging them in superposition to form an overcomplete basis (Elhage et al., 2022), at the cost of allowing interference between features. This framing has been influential, motivating dictionary learning approaches such as sparse autoencoders (SAEs), which can successfully recover interpretable features from neural representations (Templeton et al., 2024; Gao et al., 2025).

While methods for finding features continue to advance, our understanding of how feature representations arrange themselves geometrically within high-dimensional activation spaces remains limited. The geometry of features in superposition determines which concepts interfere with each other, making it a key problem in MI (Sharkey et al., 2025) with implications for SAE training (Hindupur et al., 2025), knowledge editing (Nishi et al., 2025), and adversarial robustness (Stevinson et al., 2025; Gorton & Lewis, 2025). In existing toy models, superposition is typically understood as introducing interference that should be minimized geometrically and filtered out with non-linearities such as ReLUs (Elhage et al., 2022). This gives rise to arrangements such as regular polytopes, where pairwise dot products are small and negative interference can be suppressed.

However, this perspective does not account for the kinds of structure observed in real language models. Rather than local regular polytope structure, researchers have found ordered circles of features, such as the months of the year (Engels et al., 2025), as well as anisotropic superposition, where related features cluster together rather than minimizing dot products (Bricken et al., 2023; Templeton et al., 2024). We argue that this discrepancy arises because realistic features are not sparse and uncorrelated. When features are correlated, interference need not be purely harmful: it can also be constructive,

---

[*]Joint Senior Authors

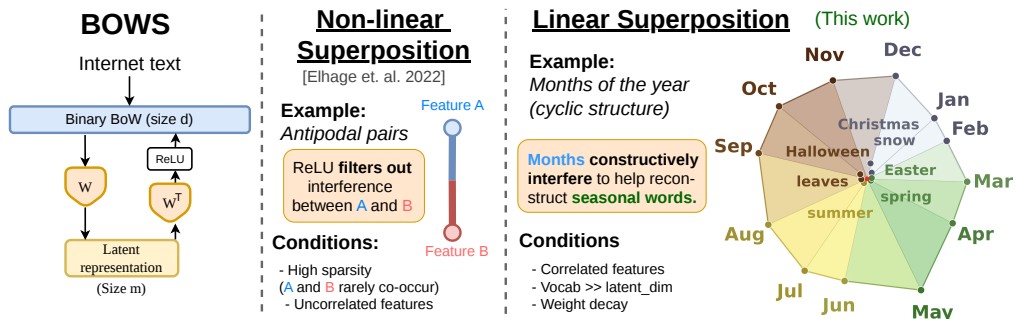

Figure 1: **BOWS, our new framework to study superposition in realistic data** (left) extends our current understanding of superposition (middle) by showing that interference can be constructive, allowing words like 'December' to contribute to the reconstruction of correlated words like 'Christmas' giving rise to a circular arrangement for the months of the year (right).

enabling efficient reconstruction in terms of weight norm and rank rather than always requiring non-linear filtering. ReLU-based filtering remains important for suppressing harmful interference, but correlated features can also be arranged so that interference reinforces signal.

To study this in a controlled setting, we introduce Bag-of-Words Superposition (BOWS), a framework in which an autoencoder is trained to encode binary bag-of-words representations of internet text in superposition. BOWS provides realistic feature correlations together with known ground-truth features. Using this setup, we show that arranging features according to their co-activation patterns naturally gives rise to semantic clusters and cyclical structures of the kind observed in real language models. We refer to the regime in which low-rank structure in the data supports constructive interference as *linear superposition*.

Our main contributions are as follows:

- We introduce BOWS as a controlled setting to study superposition with realistic features.
- We show that, when features are correlated, interference can be constructive rather than only acting as noise, and formalize a regime of *linear superposition* in which this constructive interference is leveraged by non-linear AEs to enable efficient reconstruction in terms of weight norm and rank.
- We show that these solutions emerge prominently under tight bottlenecks or weight decay, and that they reproduce key geometric structures observed in real language models, including semantic clusters and cyclical structure.
- We introduce the distinction between presence-coding and value-coding features to explain the existence of structured representations in the absence of feature correlations.

## 2 BACKGROUND

We introduce definitions that distinguish whether superposed features can be recovered by a linear decoder or require a non-linear decoder, and present our setting for studying superposition under realistic data distributions. We define superposition for abstract features $\mathbf{f}$, then discuss the relevance of superposition when these features are properties of data under the linear representation hypothesis.

### 2.1 DEFINITIONS

Consider $d$ features indexed by $[d] = \{1, \ldots, d\}$ with values $\mathbf{f} = [f_1, \ldots, f_d]^\top \in \mathbb{R}^d$ drawn from a distribution $\mathcal{D}_{\mathbf{f}}$. We consider linear encoders $\mathbf{W} \in \mathbb{R}^{m \times d}$ with $m < d$, which represent each feature along a direction $\mathbf{w}_i \in \mathbb{R}^m$.

Given a decoder $\psi : \mathbb{R}^m \to \mathbb{R}^d$, we define the per-feature coefficient of determination as

$$R_i^2(\mathbf{W}, \psi) = 1 - \frac{\mathbb{E}_{\mathcal{D}_{\mathbf{f}}}\left[(f_i - \psi(\mathbf{W}\mathbf{f})_i)^2\right]}{\mathrm{Var}_{\mathcal{D}_{\mathbf{f}}}[f_i]}, \tag{1}$$

**Definition 1** (Superposition). A set of features $F \subseteq [d]$ is represented *in superposition* if:

    1. **(Interference)** For every $i \in F$, there exists $j \in F$ with $j \neq i$ such that $\langle \mathbf{w}_i, \mathbf{w}_j \rangle \neq 0$.

2. **(Recoverability)** There exists a decoder $\psi : \mathbb{R}^m \to \mathbb{R}^d$ such that $R_i^2(\mathbf{W}, \psi) \geq 1 - \varepsilon$ for all $i \in F$.

**Definition 2** (Linear Superposition). Let $F$ be a set of features in superposition. A feature $i \in F$ is in *linear superposition* if there exists a linear decoder $\psi_{\text{lin}} : \mathbb{R}^m \to \mathbb{R}^d$ such that $R_i^2(W, \psi_{\text{lin}}) \geq 1 - \varepsilon$.

**Definition 3** (Non-linear Superposition). A feature $i \in F$ is in *non-linear superposition* if:

1. There exists a decoder $\psi$ with $R_i^2(\mathbf{W}, \psi) \geq 1 - \varepsilon$, and

2. For all linear decoders $\psi_{\text{lin}}$, we have $R_i^2(\mathbf{W}, \psi_{\text{lin}}) < 1 - \varepsilon$.

**Superposition in deep learning models**. The definitions above apply to any features $\mathbf{f}$. However, superposition becomes central to mechanistic interpretability under the *linear representation hypothesis* (LRH). The LRH reflects the empirical finding that high-level concepts such as language (Gurnee et al., 2023), entity attributes, or specific landmarks (Templeton et al., 2024) are often linearly represented in model activations. Concretely, let $\mathcal{D}_{\mathbf{x}}$ be a distribution over data samples $\mathbf{x} \in \mathcal{X}$ (e.g., an image or text) and let $\rho_1, \ldots, \rho_d : \mathcal{X} \to \mathbb{R}$ be interpretable properties (e.g., "is written in French", "contains a dog"). These induce a vector of features $\mathbf{f}(\mathbf{x}) = [\rho_1(\mathbf{x}), \ldots, \rho_d(\mathbf{x})]^\top$.

**Definition 4** (**Linear Representation Hypothesis**). We say that a hidden representation $h$ **satisfies the linear representation hypothesis** with respect to $\{\rho_j\}_{j=1}^d$ if there exist directions $\mathbf{w}_1, \ldots, \mathbf{w}_d \in \mathbb{R}^m$ such that:

$$h(\mathbf{x}) \approx \sum_{j=1}^{d} \rho_j(\mathbf{x}) \mathbf{w}_j \quad \text{for } \mathbf{x} \sim \mathcal{D}_{\mathbf{x}}. \tag{2}$$

A model that linearly represents more concepts than its hidden dimension must encode them in superposition for downstream use. Under the LRH, understanding how these concepts are geometrically arranged is thus a key challenge for mechanistic interpretability.

## 2.2 BOWS: REALISTIC DATA IN SUPERPOSITION

Studying superposition in the hidden representations of deep learning models requires postulating a set of features $\rho(\mathbf{x})$ that are linearly represented. In most cases, however, we do not have access to ground-truth features or their representations. We therefore introduce Bag-of-Words Superposition (BOWS), a setting for studying superposition with realistic feature correlations. Relative to the iid. and pairwise-correlation settings considered in Elhage et al. (2022), BOWS induces richer covariance structure, including approximately low-rank structure, while retaining known ground-truth features.

**Dataset**. Let $\mathcal{C}$ be a corpus of text segmented into *records* (lines or paragraphs). After word-level tokenisation, we construct a vocabulary of the $V$ most frequent words, discarding common English stop-words and prepositions. This vocabulary includes words such as *sun*, *code*, and *January* which often correspond to linear features in sparse autoencoders trained on language data (Engels et al., 2025; Bricken et al., 2023). Each record is then encoded as a binary bag-of-words vector $\mathbf{x} \in \{0,1\}^V$ whose $j$-th component is 1 iff the $j$-th vocabulary word appears in the record. We choose a *context size* $c \in \mathbb{N}$. For every contiguous block of $c$ records we take the element-wise logical OR of their individual vectors, obtaining a single sample. The resulting dataset is

$$\mathcal{D} = \{\mathbf{x}_i\}_{i=1}^N, \qquad \mathbf{x}_i \in \{0,1\}^V, \tag{3}$$

where $N$ is the number of $c$-record chunks in the corpus.

Experiments in the main text use WikiText-103 (Merity et al., 2017) with $V = 10,000$ and $c = 20$. Data is split into $N = 1{,}621{,}198$ samples for training, and 180,133 for validation. We include replication of the main results using OpenWebText in Appendix E.

**Autoencoder**. We use the autoencoder setup for superposition introduced in Elhage et al. (2022), consisting of an encoder with weights $\mathbf{W} \in \mathbb{R}^{m \times V}$ and bias $\mathbf{b} \in \mathbb{R}^V$, where the input $\mathbf{f} \in \mathbb{R}^V$ is reconstructed using a ReLU AE with loss: $\mathcal{L}_{\text{ReLU-AE}}(\mathbf{x}, \mathbf{W}, \mathbf{b}) = ||\mathbf{f} - \text{ReLU}(\mathbf{W}^T\mathbf{W}\mathbf{f} + \mathbf{b})||_2^2$. We also use a Linear AE as a baseline with loss: $\mathcal{L}_{\text{Linear-AE}}(\mathbf{f}, \mathbf{W}, \mathbf{b}) = ||\mathbf{f} - (\mathbf{W}^T\mathbf{W}\mathbf{f} + \mathbf{b})||_2^2$

## 3 CONSTRUCTIVE INTERFERENCE IN NON-LINEAR AES

We now study how data covariance structure and optimization constraints shape the solutions learned by linear and non-linear autoencoders. We show that when features are correlated, non-linear

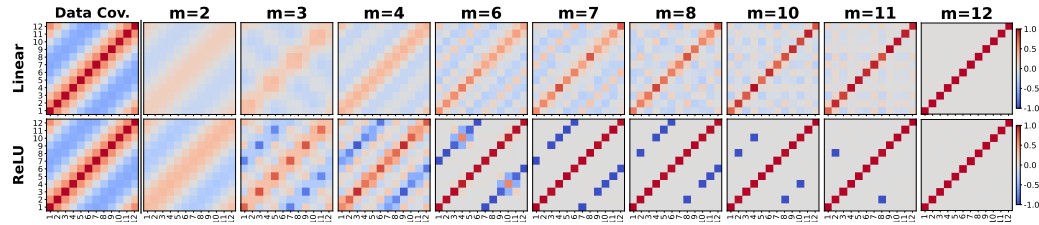

Figure 2: **Autoencoding synthetic correlated features shows two ways of handling interference.**
Weight inner products ($\mathbf{W}^\top \mathbf{W}$) at convergence for AEs encoding $d = 12$ features with cyclic
covariance, varying latent size $m$. **Top (Linear AE):** captures the top-$m$ principal components
projecting all 12 features on the circular structure induced by the data covariance. **Bottom (ReLU
AE):** Matches linear AE for small $m$, but forms antipodal pairs for $m \in \{6, ..., 10\}$ using the $ReLU$
to filter interference.

autoencoders can exploit interference constructively, arranging features so that shared variance
supports reconstruction rather than treating interference as noise to be filtered out. We characterize
when each mechanism is favored, and how the two can coexist in the same model.

## 3.1 Superposition and interference

For tied-weight AEs with weight matrix $\mathbf{W} \in \mathbb{R}^{m \times d}$, columns $\{\mathbf{w}_i\}_{i=1}^d$, and activation function $\sigma$,
the reconstruction of feature $f_i$ decomposes into a signal and an *interference* term:

$$\hat{f}_i = \sigma\Big( \underbrace{\|\mathbf{w}_i\|^2 f_i}_{\text{Signal}} + \underbrace{\sum_{j \neq i} \langle \mathbf{w}_i, \mathbf{w}_j \rangle f_j}_{\text{Interference } \mathcal{I}_i} + b_i \Big). \tag{4}$$

Let $\mathbf{\Sigma} = \mathbb{E}[\mathbf{f}\mathbf{f}^\top]$ denote the feature covariance. In the standard picture of superposition, $\mathcal{I}_i$ is treated
as noise to be non-linearly filtered out (Elhage et al., 2022). We show that this interpretation is
incomplete: when $\mathbf{\Sigma}$ has sufficiently strong low-rank structure, interference can align with the signal
and become useful rather than purely harmful.[1]

**Interference as noise (weakly correlated, sparse features).** When features are sparse and weakly
correlated, $\mathcal{I}_i$ behaves as unstructured noise that is approximately uncorrelated with $f_i$. Accurate
reconstruction then requires filtering this interference via a non-linearity (e.g. $\sigma = \text{ReLU}$) together
with a negative bias such that $\mathbb{E}[\text{ReLU}(\mathcal{I}_i + b_i)] \approx 0$, while maintaining $\|\mathbf{w}_i\|^2 \approx 1$ to preserve
signal. This is the setting emphasized in prior work, and it favors feature arrangements that minimize
pairwise dot products.

**Constructive interference** ($\text{rank}(\mathbf{\Sigma}) \leq m$). For linear AEs ($\sigma = id$)[2], the optimal map $\mathbf{P} = \mathbf{W}^\top \mathbf{W}$
is the orthogonal projector onto the top-$m$ principal components of $\mathbf{\Sigma}$ (Baldi & Hornik, 1989; Jolliffe,
2002). Defining the reconstruction residual $\varepsilon_i = f_i - \hat{f}_i$, we can rearrange Equation (4) to isolate
interference:

$$\mathcal{I}_i = (1 - \|\mathbf{w}_i\|^2) f_i - \varepsilon_i. \tag{5}$$

When $\text{rank}(\mathbf{\Sigma}) \leq m$, the data lies in the principal subspace, so $\varepsilon = 0$ and $\mathcal{I}_i = (1 - P_{ii}) f_i$. In
this case, interference is proportional to the signal rather than opposed to it. This occurs because
$P_{ij} = \langle \mathbf{w}_i, \mathbf{w}_j \rangle$ reflects the correlation between features $i$ and $j$ within the principal subspace: each
$f_j$ contributes to the reconstruction of $f_i$ in proportion to their shared variance. Through this lens,
PCA can be viewed as a form of superposition in which correlated features are arranged so that
interference reinforces signal rather than requiring suppression.[3]

**The case of real data ($\mathbf{\Sigma}$ approximately low-rank).** In real-world data, including the text data
we consider, the covariance is often well approximated by a small number of principal components

---

[1]Elhage et al. (2022) also consider pairwise correlated features, but in their setup all principal components
carry significant variance, so PCA collapses correlated pairs onto indistinguishable points.

[2]In the linear case, the optimal bias satisfies $b^\star = (I - \mathbf{W}^\top \mathbf{W})\mathbb{E}[f]$, so the AE effectively operates on
centered features $f - \mathbb{E}[f]$. We therefore analyse the setting with $b = 0$ when $\sigma = id$ without loss of generality.

[3]In practice, $P$ is learned from finite data and reflects the sample covariance; for test samples whose
correlations deviate from the training distribution, interference will be imperfectly aligned with the signal.

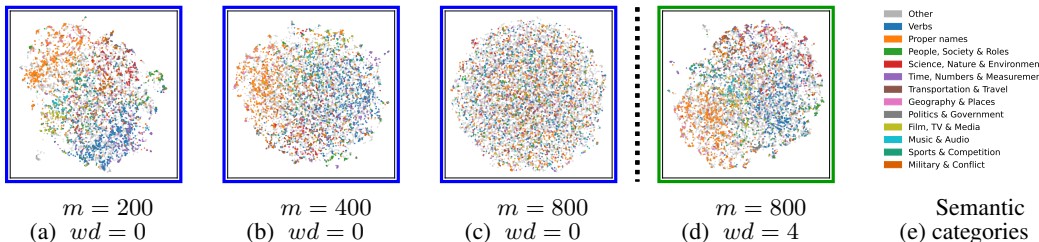

$$\begin{array}{ccccc} m = 200 & m = 400 & m = 800 & m = 800 & \text{Semantic} \\ \text{(a)} \;\; wd = 0 & \text{(b)} \;\; wd = 0 & \text{(c)} \;\; wd = 0 & \text{(d)} \;\; wd = 4 & \text{categories} \\ & & & & \text{(e)} \end{array}$$

Figure 3: **Linear superposition appears in ReLU AEs which have small latent sizes (a) or are trained with weight decay (d), giving rise to semantic clusters.** UMAP projections of word embeddings from AEs of different latent dimensions ($m$) and weight decay values ($wd$). Points are colored by semantic category (e).

(Deerwester et al., 1990; Blei et al., 2003; Udell & Townsend, 2019), making $\Sigma$ approximately low-rank. In this regime, interference behaves as signal contaminated by a residual $\varepsilon_i$ corresponding to variance outside the top-$m$ subspace, with $\|\varepsilon\|^2 = \sum_{k>m} \lambda_k(\Sigma) \geq \min_{\text{rank}(\widehat{\Sigma}) \leq m} \|\Sigma - \widehat{\Sigma}\|_F$ (Eckart & Young, 1936). Thus, as the spectrum of $\Sigma$ becomes more concentrated, the residual becomes smaller and constructive interference becomes increasingly effective.

**When do non-linear models exploit constructive interference?** For models trained on real data with weight decay, the interference-filtering and constructive-interference solutions have different norm requirements. In the standard sparse-feature setting, recovering each feature while relying on a ReLU to suppress interference requires feature directions with approximately unit norm, so the canonical interference-filtering solution has $\|\mathbf{W}\|_F^2$ scaling with the number of represented features, i.e. $\|\mathbf{W}\|_F^2 \approx d$. By contrast, when reconstruction is achieved by projecting onto an $m$-dimensional low-rank subspace, the optimal linear solution is a rank-$m$ projector $P = \mathbf{W}^\top \mathbf{W}$, for which

$$\|\mathbf{W}\|_F^2 = \text{tr}(P) = \sum_k \lambda_k(P) = m < d. \tag{6}$$

Thus, *weight decay combined with tight bottlenecks* ($m \ll d$) biases even non-linear models toward solutions that exploit low-rank structure, since these can achieve accurate reconstruction with substantially smaller weight norm.

**Complementarity of the two mechanisms**. The constructive and filtering mechanisms are not mutually exclusive. When $\text{rank}(\Sigma) \leq m$, a linear AE can arrange features so that interference on inactive features vanishes exactly. But when $\Sigma$ is only approximately low-rank, the residual $\varepsilon_i$ can still induce false positives on inactive features or negative reconstruction values. In this setting, both mechanisms can operate together: the weight geometry exploits correlation structure so that much of the interference aligns with the signal, while a ReLU with negative bias suppresses the residual harmful interference introduced by $\varepsilon_i$.

### 3.2 Evidence of linear superposition in non-linear AEs

We start our empirical analysis with a simplified setting where two AEs of latent dimension $m$ with and without a ReLU in the decoder are trained on 12 dimensional data with a cyclic covariance structure (Figure 2 left). Details for the data generation process are provided in Appendix A. Figure 2 (top row, Linear) shows the baseline learned by the linear AE, which learns the projection onto the principal subspace (Baldi & Hornik, 1989). We then study the emergence of linear superposition in non-linear autoencoders, specifically looking at autoencoders with a ReLU in the decoder of the kind described in Section 2.2.

**Linear superposition**. When the bottleneck is very tight ($m \ll d$), the ReLU AE recovers the circular structure dictated by the top principal components Figure 2 (bottom row, ReLU, $m < 6$). These are examples of a non-linear AE leveraging linear superposition.

**Non-linear superposition**. Figure 2 (bottom row, ReLU, $m \geq 6$) shows that as $m$ increases, the ReLU AE abandons the circular PCA structure and instead represents features as *antipodal pairs*. This specific geometry is one of the cases studied by Elhage et al. (2022), whereby features are placed

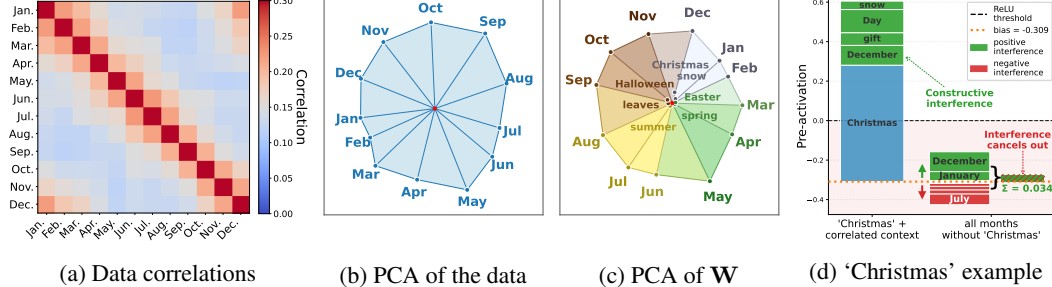

| (a) Data correlations | (b) PCA of the data | (c) PCA of $\mathbf{W}$ | (d) 'Christmas' example |

Figure 4: **Circular representation of months arises from data covariance via PCA. (a)** Empirical correlation matrix of month words in the WikiText-103 BOWS dataset, showing cyclic correlations. **(b)** PCA applied directly to the 12 month dimensions of the BOWS data vectors, projected onto the top 2 PCs, reveals a circle. **(c)** PCA applied to the 12 learned encoder features ($W$ columns) for months from a ReLU AE trained on WikiText-BOWS ($V = 10k$, $m = 1000$), projected onto their top 2 PCs, also recovers the circular structure. Seasonal words such as 'Christmas' and 'summer' align with the months with which they co-occur, allowing 'December' to contribute to the reconstruction of 'Christmas' while interference on 'Christmas' cancels out if all months are present **(d)**.

in anti-correlated pairs such that activating one feature negatively activates its antipodal partner, with negative interference zeroed out by the ReLU as an example of non-linear superposition.

**Structure disappears as features become orthogonal**. As the latent size approaches the input size, the autoencoder weights converge to the identity, representing each feature orthogonally for a perfect reconstruction. As these models ($m = 12$) approach a perfect reconstruction of the data, the circular covariance structure is no longer reflected in the weights, and any notion of superposition is lost.

## 4 LINEAR SUPERPOSITION EXPLAINS FEATURE GEOMETRY IN REAL DATA

In Section 3, we argued that when features are correlated, interference need not be purely harmful: it can also be constructive, allowing non-linear autoencoders to exploit low-rank structure in the data rather than relying only on ReLU-based interference filtering. In this section, we show that this mechanism accounts for the feature geometry observed in our main WikiText-BOWS setting.

**Semantic clustering from constructive interference**. A prominent observation in studies of LLM activations is that learned features often form clusters based on semantic relatedness, leading to *anisotropic superposition*, where features are not arranged to minimize pairwise dot products (Bricken et al., 2023; Templeton et al., 2024). Such structure is difficult to reconcile with the standard picture of superposition as purely harmful interference. By contrast, it arises naturally if a model exploits constructive interference to capture low-rank structure in realistic data.

In Figure 3 we show UMAP (McInnes et al., 2018) projections of the word embeddings (columns of $\mathbf{W}$) learned by a ReLU AE trained on WikiText-BOWS ($V = 10,000$) with varying latent dimensions $m$. In panel (a), where compression is strong ($m = 200$), we observe distinct clusters corresponding to semantic categories (e.g. verbs, proper names, sports). This clustering weakens as the latent size increases to $m = 800$ (Figure 3c), but reappears when weight decay is introduced (Figure 3d). These results suggest that semantic clustering can arise as a consequence of constructive interference under compression, and help explain why similar structure is observed in language models trained with weight decay.

**Cyclical structures inherited from data statistics**. Another notable form of feature geometry observed in real models is circular structure in the principal components of feature embeddings, for concepts such as months of the year or days of the week (Engels et al., 2025).

Consider the features corresponding to the twelve months of the year. In Figure 4a, we show that these features exhibit cyclic correlations in WikiText: for example, January co-occurs more often with February and December than with August. This covariance structure drives the leading principal components of the month activations, producing a circle in the 2D PCA plot in Figure 4b. Notably, the learned latent representations exhibit the same geometry (Figure 4c). For a fully linear AE, this would

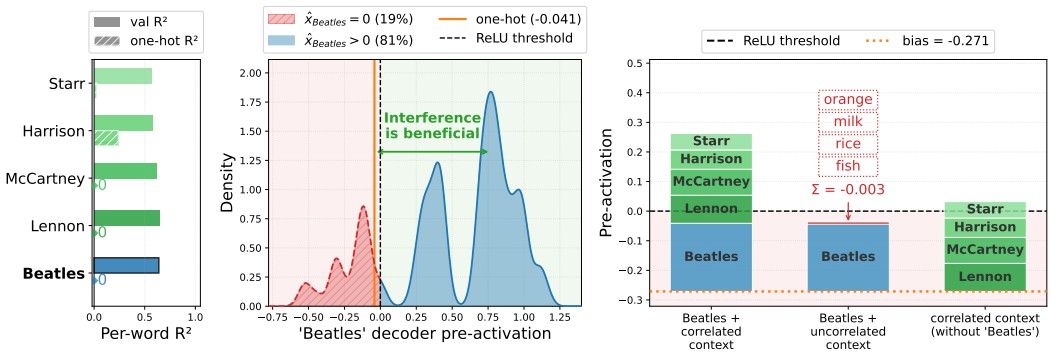

Figure 5: **Constructive interference and interference filtering coexist in realistic data. (Left)** Terms related to 'Beatles' achieve high validation $R^2$ despite poor one-hot reconstruction, indicating that they benefit from contextual interference. **(Middle)** For 81% of validation samples containing 'Beatles', interference improves reconstruction relative to the one-hot case. **(Right)** In supportive contexts, correlated words contribute positive pre-activation to 'Beatles'; when these contexts occur without the target word, the ReLU and negative bias suppress false positives.

be expected, since its weights span the PCA subspace (Baldi & Hornik, 1989) (see Appendix D). The fact that our non-linear AE displays the same structure supports the presence of linear superposition within a non-linear model.

Furthermore, seasonal words such as 'Christmas' and 'summer' align with the months with which they co-occur. This provides a direct example of correlated features being arranged so that interference reflects shared structure in the data rather than being eliminated. We show as an example, that interference from 'December' contributes to the reconstruction of 'Christmas' when they are both present in the context, while interference from all the months almost cancels out and does not lead to false positives when 'Christmas' is not present (Section 4). More detailed results for 'Christmas' and other examples where interference is constructive are provided in Appendix C.

To verify that the month features are in linear superposition, we train a linear decoder (without a ReLU) to reconstruct their inputs. We obtain $R_i^2(\mathbf{W}_{\text{months}}, \psi_{\text{lin}}) = 0.98 \pm 0.00015$, with none of the months orthogonal to one another. By Definition 2, the month features are therefore represented in linear superposition.

**Constructive interference and interference filtering coexist in realistic data**. In realistic data, constructive interference and ReLU-based filtering can coexist within the same model, and even for the same feature. This can be seen by comparing reconstruction in isolation and in context. If interference were purely harmful, a feature would be reconstructed best in the one-hot setting, where no other active features contribute interference. However, when a model exploits correlations between features, the one-hot setting need not be optimal: correlated context can improve reconstruction by contributing constructively to the target feature.

We observe exactly this phenomenon for words associated with The Beatles. These words are not frequent enough to occupy nearly independent dimensions, yet 'Beatles' and the surnames of the band members achieve validation $R^2$ values above $0.5$ while having near-zero one-hot $R^2$. Thus, interference from correlated words is not merely tolerated; it is beneficial. As shown in Figure 5 (middle), for 81% of validation samples containing 'Beatles', the contextual reconstruction is better than the one-hot reconstruction. Figure 5 (right) shows the mechanism more directly: correlated words provide positive evidence for the target word, while the ReLU and negative bias suppress false positives when similar contexts occur without the target. This is precisely the realistic setting in which constructive interference and interference filtering complement one another.

Word frequencies in natural text are highly heterogeneous (Zipf, 1949; Clauset et al., 2009), so different features need not be represented by the same mechanism. Some can exploit correlation structure and remain in linear superposition, while others are represented through stronger interference filtering or become nearly orthogonal. Performing a linear superposition test on the $m = 800$ model used in Figure 9, and using $R = 0.5$ as a threshold, we find that 4073 words are represented in non-linear superposition, while the most frequent 522 words are represented in linear superposition.

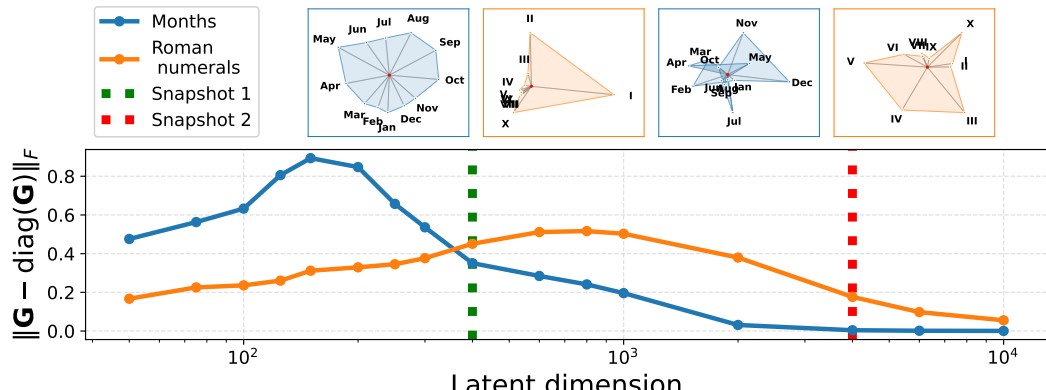

Figure 6: **Different feature structures disappear at different latent sizes.** As the latent size of the AE increases, the structure in the weights appears and disappears at different times for different groups of features. We look at 2D PCA plots of the months and roman numeral embeddings at two key latent sizes. At Snapshot 1 (green line), both groups of features appear in order with some structure, while at Snapshot 2 (red line), the representations of the months have already become orthogonal while the roman numerals are still represented in an ordered structure.

We illustrate this heterogeneity in Figure 6 by comparing two groups of words: the months of the year and the first 10 Roman numerals. We isolate the weights $\mathbf{W}_{\text{months}}$ and $\mathbf{W}_{\text{roman}}$ and measure the off-diagonal Frobenius norms of $\mathbf{W}_{\text{months}}^T \mathbf{W}_{\text{months}}$ and $\mathbf{W}_{\text{roman}}^T \mathbf{W}_{\text{roman}}$, which quantify the extent to which features in each group interfere with one another. An off-diagonal Frobenius norm of zero indicates that all features in the group are mutually orthogonal. Figure 6 shows that at small latent sizes both groups exhibit ordered structure consistent with their covariance. As the latent size increases, however, the month structure disappears earlier, as $\mathbf{W}_{\text{months}}$ becomes closer to orthogonal while the Roman numerals continue to display ordered structure. This illustrates that in realistic data, where features differ in frequency and correlation structure, different groups of features can occupy different parts of the spectrum between linear and non-linear superposition.

## 5 VALUE-CODING FEATURES: GEOMETRY WITHOUT INPUT CORRELATIONS

While the BOWS framework can replicate the kind of semantic structure observed in the hidden representations of language models, there are some examples, like the circles that appear in models performing modular addition (Power et al., 2022; Nanda et al., 2023), which appear in the absence of correlations in the data. To explain this kind of structure, we introduce the distinction between *value-coding* and *presence-coding* features and explain how value-coding features can give rise to apparent structures in features that are not actually represented in superposition.

**Presence-coding features**. We say that a representation $h(x) \in \mathbb{R}^d$ contains a *presence-coding feature* if some binary variable $y(x)$ (e.g. "this token is the word *cat*") is recoverable by a linear classifier. Formally, there exist weights $\{w_k, b_k\}$ such that $\hat{y}(x) = \arg\max_k(w_k^\top h(x) + b_k)$ predicts $y(x)$ with low error. Presence-coding features thus behave as detectors for discrete properties, and different values of $y$ are treated as separate classes without requiring any particular geometric relation between them in representation space *a priori*. For presence-coding features structured representations are contingent on correlations in the data and capacity constraints that lead the features to be represented in linear superposition.

**Value-coding features**. In contrast, we say that a representation $h(x)$ contains a *value-coding feature* if a real-valued variable $v(x) \in \mathbb{R}$ (e.g. an angle, a coordinate, or a continuous latent factor) is linearly decodable. That is, there exist $w \in \mathbb{R}^d$ and $b \in \mathbb{R}$ such that $\hat{v}(x) = w^\top h(x) + b$ approximates $v(x)$ with low error. A collection of such value-coding features $v_1(x), \ldots, v_k(x)$ defines a low-dimensional *value space* $\mathbb{R}^k$; plotting examples in this space can reveal semantically meaningful structures (for instance, a 2D map from $(x, y)$ coordinates, or a circle from $(\sin\theta, \cos\theta)$ in modular addition). Crucially, these structures are fully accounted for by the existence of linear value codes for the underlying variables and therefore exist even in the absence of superposition.

**Empirical evidence for value-coding features**. To exemplify this, we look at a simplified modular addition setup similar to that in Nanda et al. (2023), as well as a relative map position setup inspired by Gurnee & Tegmark (2024). In the latter, we take the top 1,000 most populated cities in the US and calculate their relative positions in terms of quadrants (e.g. Seattle is north-west of Denver). The cities are embedded separately, fed through a 1-hidden layer ReLU MLP (details in Appendix A). This latter dataset is designed to incentivize the model to learn 2 value-coding features encoding the coordinates of each city since relative positions of the cities can easily be calculated by subtracting coordinates. For both of these tasks, each integer or city pair only appears once across the train and validation sets so no pair of cities or integers are correlated and we do not expect any kind of low rank structure coming from linear superposition.

We validate that the cardinal direction model is learning value coding features for the coordinates by training a linear probe to predict the coordinates of a subset of the cities from their embeddings. We then use this probe to predict the coordinates of some held-out cities. The result is an average $R^2$ validation score of 0.98 and a correct arrangement of the held out cities on the US map when projected onto the directions identified by the linear probes (Figure 7). Similarly, we validate that the relevant sine and cosine values are linearly represented by projecting the representations onto the corresponding Fourier components in the case of modular addition (details in Appendix A).

We have shown two examples where neural networks need linearly represent values like coordinates or trigonometric functions, in order to perform computations with them. We have then shown in Figure 7 that in doing this, models implicitly create structures like circles or maps when we project the data onto these value-coding features.

**Ablating the subspace orthogonal to the value-coding features.** Having isolated the value coding features both in the modular addition and map datasets, we ablate the subspace of the embedding space that is orthogonal to these value-coding features. In both cases, the ablation results show that most of the test accuracy is preserved if we replace over 90% of the dimensions with their mean (Table 1). Conversely performance breaks down if we remove only the value coding features. This serves as further evidence that these value coding features are the units of computation that the model is using to perform these tasks (replicating the results from Nanda et al. (2023) in the modular case).

Table 1: Value-coding (VC) ablations on two datasets. VC$^+$ keeps the VC subspace and zeros its orthogonal complement; VC$^-$ ablates VC coordinates.

| Condition | MAP | | Key-freq | |
|---|---|---|---|---|
| | Loss ↓ | Acc. (%) ↑ | Loss ↓ | Acc. (%) ↑ |
| Baseline | 0.0536 | 97.94 | 0.0001 | 100.00 |
| VC$^+$ | 0.2879 | 93.16 | 0.1649 | 93.99 |
| VC$^-$ | 6.3943 | 22.43 | 11.7464 | 3.11 |

**Distinguishing feature geometry and feature manifolds**. At first glance, feature manifolds such as those in Figure 7 may appear deceptively similar to geometric arrangements like Figure 4. Yet, our findings offer a principled way to distinguish the two. When features exhibit a recognizable structure, we can ask whether that structure reflects genuine co-activation patterns. In Figure 4, for instance, the latent arrangement clearly aligns with co-activations among month-related features, pointing to a superposition-based representation. In contrast, the patterns in Figure 7 emerge despite inputs being uncorrelated, suggesting the model has instead learned value-coding features through task-driven projections. Disentangling these phenomena in more complex, real-world scenarios remains a compelling direction for future research.

# 6 RELATED WORK

**Superposition**. Initial works in MI studied interpretable monosemantic neurons in DL models (Olah et al., 2020; Cammarata et al., 2020) but faced challenges in interpreting polysemantic neurons which activate for seemingly unrelated concepts. Elhage et al. (2022) introduced superposition as an explanation for neuron polysemanticity. This view of DL models inspired further studies (Scherlis et al., 2025) and sparse dictionary learning approaches like sparse autoencoders to decompose model activations into an overcomplete basis of linear features (Gurnee et al., 2023; Huben et al., 2024; Bricken et al., 2023). This approach has successfully been scaled to frontier language models and multimodal models by Gao et al. (2025) and Templeton et al. (2024).

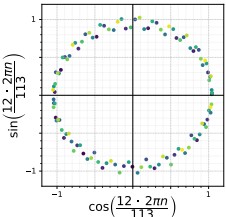 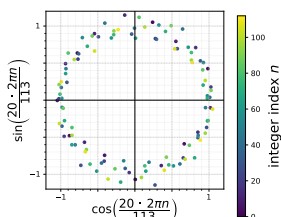 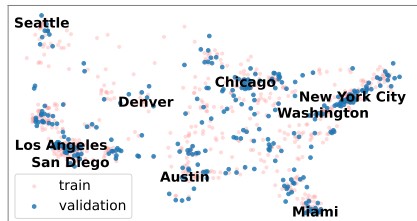

Figure 7: In the embeddings of a model performing modular addition, the circular structure is isolated by projecting onto directions corresponding to sine and cosine values (**left**). In the case of the map directions dataset, two linear probes reconstruct the city coordinates, reconstructing their positions on the map for both train and validation samples. (**right**).

**Feature geometry**. Park et al. (2024) proposed a formalization of the LRH and proposed an inner product that preserves language structure. Park et al. (2025) studied how features with hierarchical relations are encoded in language models while they show that categorical features which form polytopes, we note that these are different from *regular* polytopes posited by Elhage et al. (2022). Lee et al. (2025) studied geometric similarities in token embeddings of different language models, while Zhao et al. (2024) studied the kind of structure that emerges in the representations of models trained on next token prediction. Li et al. (2025) showed that language models represent integers in a helix structure to perform modular addition echoing the results from Nanda et al. (2023) and Liu et al. (2022) on transformers trained for modular addition. Gurnee & Tegmark (2024) showed that longitude and latitude as well as a notion of time, are encoded as linear features in language models. The main results highlighted in this paper are circular structures formed by features and semantic feature clusters, described in Engels et al. (2025) and Bricken et al. (2023) respectively. These structures have also been studied as feature manifolds in Modell et al. (2025), and Hindupur et al. (2025) highlighted the importance of understanding feature geometry when designing SAEs. These findings sparked a discussion around the potential limitations of SDL approaches and the LRH suggested by this non-linearly encoded semantic information (Sharkey et al., 2025). Understanding feature geometry has implications beyond interpretability and generalization, such as for robustness and compression (Stevinson et al., 2025; Barsbey et al., 2026).

**Structure in word representations**. Classic work on distributional semantics and word embeddings (e.g., Word2Vec (Mikolov et al., 2013), GloVe (Pennington et al., 2014)) demonstrated that training simple models on large text corpora leads to vector spaces where geometric relationships capture surprisingly sophisticated semantic and syntactic relationships. Levy & Goldberg (2014) showed that methods like Word2Vec with negative sampling implicitly factorize the Pointwise Mutual Information (PMI) matrix shifted by a constant, while others show connections to PCA or SVD on co-occurrence counts or PMI (Allen & Hospedales, 2019).

# 7    DISCUSSION & CONCLUSION

We introduced BOWS as a controlled setting for studying superposition with realistic feature correlations. Our main result is that, when features are correlated, superposition does not only introduce harmful interference to be filtered out: it can also exploit constructive interference, arranging features according to their co-activation patterns. This mechanism yields norm- and rank-efficient reconstructions and accounts for semantic clusters and cyclical structures of the kind observed in real language models (Bricken et al., 2023; Engels et al., 2025). We show this both in synthetic settings and in realistic internet text, and find that such solutions are especially prevalent under tight bottlenecks and weight decay. Finally, we distinguish these correlation-driven structures from value-coding feature manifolds, which arise for functional reasons.

**Limitations and future work**. BOWS is intentionally simple, and does not capture the full richness of language model representations. Our results show that constructive interference and ReLU-based interference filtering can coexist, but we do not yet provide a complete mathematical characterization of when each mechanism dominates. Important directions for future work include extending the analysis to untied autoencoders and more realistic representation settings, and using BOWS as a benchmark for SAE evaluation with known ground-truth feature geometry.

**Acknowledgments**. This work was supported by the UKRI Centre for Doctoral Training in Safe and Trusted AI [EP/S0233356/1]. TB acknowledges support from the Engineering and Physical Sciences Research Council [grant EP/X011364/1]. TB was supported by a UKRI Future Leaders Fellowship [grant number MR/Y018818/1]. MB was supported by the EPSRC Project GNOMON [EP/X011364/1].

## LLM USAGE STATEMENT

This work used LLM assistance for literature search, language editing, coding, and an initial semantic grouping of 4,000 words that was subsequently inspected and refined by hand. All LLM-assisted code and writing were validated by the authors, who take full responsibility for the final manuscript and released code.

## REPRODUCIBILITY STATEMENT

We include code to reproduce the main results of this paper (including Figure 2, Figure 3 and Figure 4) in the supplementary material. Details about the BOWS setup are given in Section 2.2 and further details about our experiments are provided in appendix A.

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

## APPENDIX

## A   IMPLEMENTATION DETAILS

**WikiText-BOWS**. All the models trained in the WikiText BOWS setup use a cosine annealing scheduler with a starting learning rate of $1e - 3$ and are trained for 20 epochs with a batch size of 1024.

**Synthetic "Months" dataset.** Each document is a 12-bit vector $x \in \{0, 1\}^{12}$ whose entries stand for the calendar months. One sample is generated as follows.

1. *Latent month angle.* Pick a discrete month $m \in \{0, \dots, 11\}$ (uniformly or by cycling) and add Gaussian blur:
$$\theta = 2\pi m/12 \; + \; \varepsilon, \qquad \varepsilon \sim \mathcal{N}(0, \sigma_\theta^2).$$

2. *Embed on the unit circle.* $z = \begin{bmatrix} \cos\theta, \; \sin\theta \end{bmatrix}^\top \in \mathbb{R}^2$.

3. *Project onto month directions.* Let
$$W = \left[ (\cos \tfrac{2\pi k}{12}, \; \sin \tfrac{2\pi k}{12}) \right]_{k=0}^{11} \in \mathbb{R}^{12 \times 2},$$
whose $k$-th row corresponds to month $k$. Compute log-odds $\ell_k = \beta W_k z + b$, where $b < 0$ fixes the global sparsity and $\beta > 0$ controls sharpness.

4. *Binary activations.* Draw the bits independently:
$$x_k \; \sim \; \text{Bernoulli}\big(\sigma(\ell_k)\big), \qquad \sigma(u) = \tfrac{1}{1+e^{-u}}, \qquad k = 1, \dots, 12.$$

With $\sigma_\theta{=}0$ and large $\beta$ the code is nearly one-hot; decreasing $\beta$ or increasing $\sigma_\theta$ mixes neighboring months, producing a rank-2 correlation structure that is analytically tractable yet retains the extreme sparsity of real bag-of-words data.

**Modular addition**. Let $a, b \in 0, \dots, 112$ make an input pair of integers with the task being addition modulo 113. We learn a shared embedding matrix $E \in \mathbb{R}^{113 \times 100}$ that maps each integer to a 100-dimensional vector. For a given pair, we look up $E[a]$ and $E[b]$, concatenate them into a 200-dimensional feature vector, and feed this through a three-hidden-layer MLP. Each hidden layer has 200 ReLU neurons. The output layer produces logits over the 113 possible sums. We train the entire model end-to-end via cross-entropy loss using the AdamW optimizer with weight decay set to 4.

**Relative map positions**. From the Cities1000 Dataset (GeoNames, 2025), we take the top 1,000 most populated cities in the US and sample two subsets of city pairs out of the $1,000,000$ possible pairs. For each pair of cities $a, b$, the task is to predict their relative position on the US map out of eight possible classes (North, South, East, West, North–East, North–West, South–East, South–West). We learn a 200-dimensional embedding matrix $E \in \mathbb{R}^{1000 \times 200}$ to map each city to a 50-dimensional vector; for each pair $(a, b)$, we concatenate their embeddings $E[a]$ and $E[b]$ into a 100-dimensional feature vector, which is then fed through a single hidden-layer MLP with 200 ReLU units. The MLP's output layer produces logits over the eight classes, and the entire model is trained end-to-end using cross-entropy loss and an Adam optimizer.

**UMAP plots and semantic clusters**. For the UMAP plots in Figure 1 and Figure 3, the categories are created by using Gemini 2.5 Pro to split the top 4000 words into categories, with each category inspected and refined by hand. The exact word to category mappings can be found in the code provided in the supplementary material. The UMAP plots are made with 15 neighbors, a min distance of 0.01, and a cosine metric.

**Linear probes**. In this work, we argue that non-linear AEs can sometimes linearly encode low-rank structure of the data. To quantify how *linear* the representations learned by the ReLU-AE are, we deploy a simple linear probe. After fully training the ReLU-AE, we freeze its encoder and collect the latent activations $\mathbf{h} = \mathbf{W}\mathbf{x} \in \mathbb{R}$. A probe is a single linear layer $\mathbf{P} \in \mathbb{R}^{V \times m}$ that is trained from scratch to reconstruct the input without any non-linearity:
$$\hat{\mathbf{x}}_{\text{probe}} = \qquad \mathcal{L}_{\text{probe}}(\mathbf{x}, \mathbf{P}) = \big\| \mathbf{x} - \hat{\mathbf{x}}_{\text{probe}} \big\|_2^2. \tag{7}$$

To measure how much of the ReLU-AE's predictive power can be captured by a linear mapping we use the Fraction of Explained Variance (FEV) of the probe relative to the ReLU-AE:
$$\text{FEV} = 1 - \frac{\sum_i \big\| \hat{\mathbf{x}}_{i,\text{ReLU}} - \hat{\mathbf{x}}_{i,\text{probe}} \big\|_2^2}{\sum_i \big\| \hat{\mathbf{x}}_{i,\text{ReLU}} - \bar{\mathbf{x}}_{\text{ReLU}} \big\|_2^2}, \tag{8}$$

where $i$ indexes data points and $\bar{\mathbf{x}}_{\text{ReLU}}$ is the mean ReLU-AE reconstruction over the evaluation set. An FEV of 1 indicates that a *purely linear* map can reproduce the ReLU-AE's outputs perfectly; an FEV of 0 means the probe does no better than predicting the mean. We report the FEV on a held-out validation split.

## B  TOY TRANSFORMER SETTING

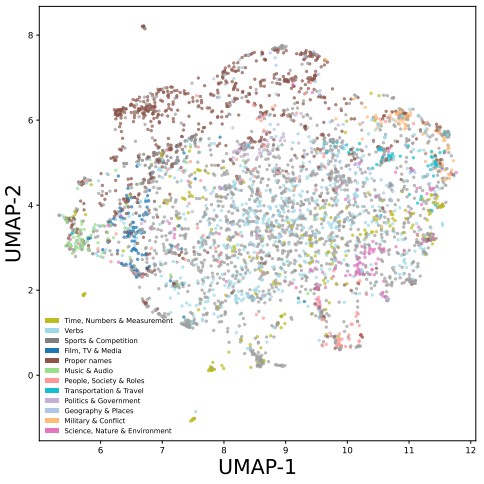
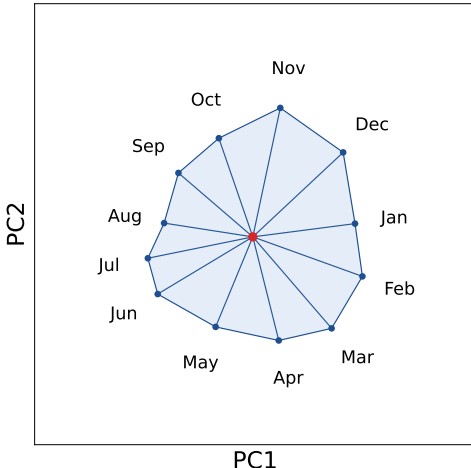

(a) UMAP of BOWS word embeddings  (b) PCA of unembedding weights

Figure 8: **Tow-layer transformer trained on Wikitext-103 for the multi-task token-recovery task exhibit semantic clusters and ordered circular representations for the months of the year.** This replicates the main results in Figure 4 and Figure 3 on a different dataset. Colors correspond to different semantic categories (Figure 3d).

We train a two-layer transformer on WikiText-103 to predict, at each position, the set of vocabulary items that has appeared so far in the causal context.

**Architecture**. One encoder block with pre-norm attention and MLP:

- Token embedding dimension $d_{\text{model}} = 768$, tied to the output projection.
- Multi-head self-attention with 8 heads, dropout 0, causal masking, and learned positional embeddings.
- Feed-forward network of width $4d_{\text{model}}$ (GELU activation), followed by layer normalization.

**Data and tokenization**. WikiText-103 is tokenized at the word level with a fixed vocabulary of 16,000 tokens. The tokenizer and reserves `<pad>` and `<unk>`. Sequences are constructed with window length 512 and stride 512, padding to full length.

**Targets**. For each sequence position, the target is a multi-hot vector over the vocabulary indicating whether the token has appeared anywhere in the prefix (inclusive). This is computed from the sequence with padding tokens masked out.

**Loss and optimization**. Training uses binary cross-entropy with logits over the multi-hot targets. Optimization uses AdamW with learning rate $3 \times 10^{-4}$, weight decay $5 \times 10^{-2}$, batch size 8, and cosine annealing schedule. Gradients are clipped to norm 1.0.

In Figure 8 we see that, similarly to the BOWS setup, semantic clusters and circular structures appear in the residual stream of the transformer as a byproduct of compression.

## C  COMBINING CONSTRUCTIVE INTERFERENCE AND INTERFERENCE FILTERING

The main text argues that realistic models need not rely on a single pure mechanism. In practice, correlated features can contribute constructively to reconstruction, while the ReLU and negative bias still suppress false positives. The examples below illustrate this coexistence for features that are only partially reconstructable in isolation.

To complement the 'Beatles' example in Figure 5, we now show the same analysis for 'Christmas'. While 'Christmas' is frequent enough to be allocated a large weight norm and achieve a positive reconstruction whenever it is present in the input, even in unrelated context, it only achieves a reconstruction of 0.2 in the one-hot case. In

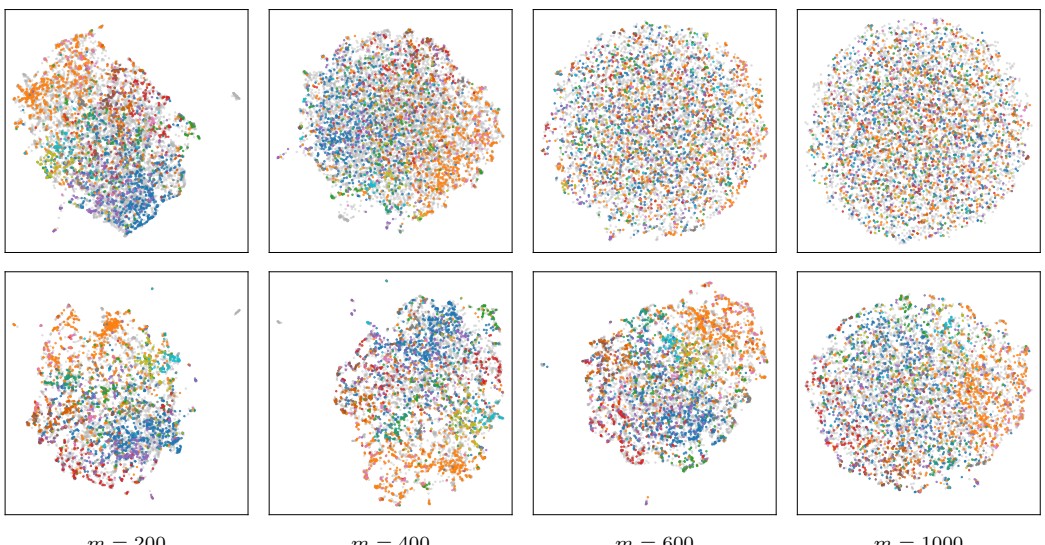

$m = 200$ $\quad\quad\quad$ $m = 400$ $\quad\quad\quad$ $m = 600$ $\quad\quad\quad$ $m = 1000$

Figure 9: UMAP embeddings of features from AEs trained with context size of 20 records with wd=0 (top) and wd=4 (bottom) across different latent sizes. The plot shows that semantic structure remains for a larger fraction of context sizes when using weight decay. Colors correspond to different semantic categories (Figure 3d).

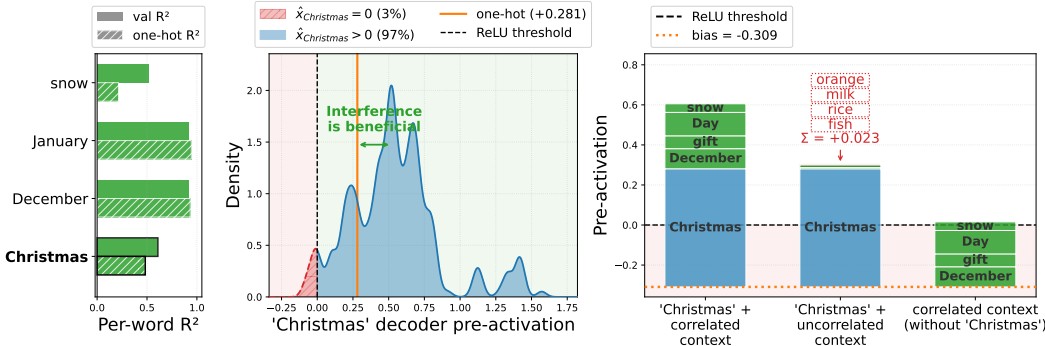

Figure 10: **Words like 'December' and 'Day' contribute to the reconstruction of 'Christmas'.** (**left**) 'Christmas' is better reconstructed with than without interference. (**middle**) For most samples (81%) where 'Christmas' appears in the validation set, interference is beneficial, allowing for reconstructions closer to 1. (**right**) When 'Christmas' appears in the right context it has a positive reconstruction thanks to the interference from related terms. However, if the related context is present but 'Christmas' is not, the model uses the ReLU and a negative bias to avoid false positives at the cost of having smaller reconstructions for 'Christmas' when it appears in an unusual context.

order to get closer to the target of 1, the reconstruction of 'Christmas' relies on words like 'December' or day Figure 11

## D   PCA & GEOMETRY PRESERVATION

We compute correlations in data by the *Pearson correlation coefficient*. A *Pearson correlation matrix* $\mathbb{R}$ is a *Gram* (inner-product) matrix of the standardized vectors. Due to the normalization, the Euclidean distance (chordal) distance monotonically links to the angular distances:

$$d_{ij}^2 = \|\mathbf{x}_i - \mathbf{x}_j\|^2 = 1 - \cos\left(\mathbf{x}_i^\top \mathbf{x}_j\right) = 2\left(1 - R_{ij}\right),$$

where $\|\mathbf{x}_i\| = 1$, and $R_{ij}$ is the $i^{\text{th}}$ and $j^{\text{th}}$ entry of $R$. So correlation induces Euclidean geometry (up to rotation/reflection) on the embedded points. Hence, performing PCA on $R$ is equivalent to performing classical *multidimensional scaling* (MDS) (Cox & Cox, 2008) on chordal distances, which explicitly aims to embed

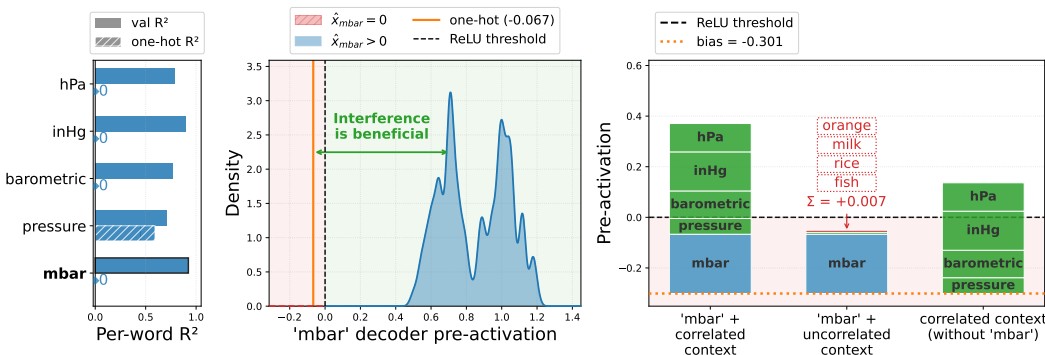

Figure 11: **Atmospheric terms like 'barometric' and 'inHg' contribute to the reconstruction of 'mbar'.** (**left**) 'mbar' is better reconstructed with than without interference. (**middle**) For every sample where 'mbar' appears in the validation set, interference is beneficial, allowing for reconstructions closer to 1. (**right**) When 'mbar' appears in the right context it has a positive reconstruction thanks to the interference from related terms. However, if the related context is present but 'mbar' is not, the model uses the ReLU and a negative bias to minimize false positives at the cost of not being able to reconstruct 'mbar' when it appears in an unusual context.

vectors such that between-vector distances are preserved as well as possible. Hence PCA yields embeddings which reflect the geometry induced by the correlations in data.

# E    OPENWEBTEXT REPLICATIONS

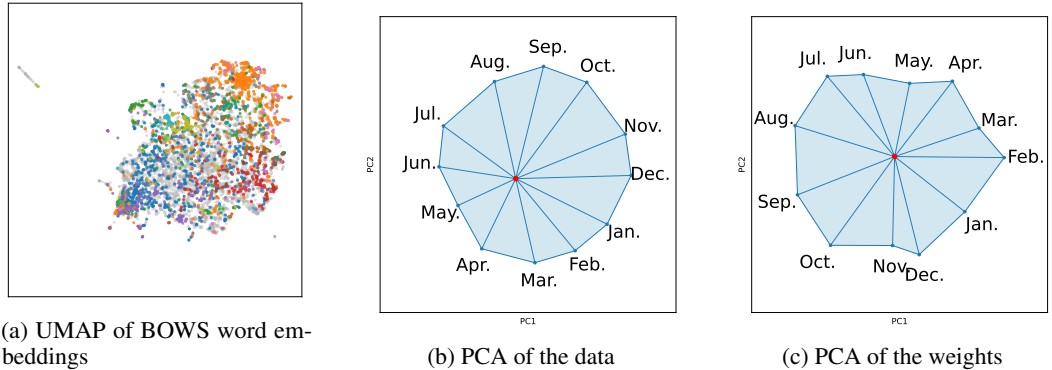

(a) UMAP of BOWS word embeddings

(b) PCA of the data

(c) PCA of the weights

Figure 12: **Autoencoders trained on OpenWebText exhibit semantic clusters and ordered circular representations for the days of the week.** This replicates the main results in Figure 4 and Figure 3 on a different dataset. Colors correspond to different semantic categories (Figure 3d).

In Figure 12 we replicate the main results of the paper showing that the appearance of circular structure for the months of the year and semantic clusters is not a phenomenon limited to WikiText. These results are from an OpenWebText BOWS setup with $v = 10,000$, $c = 10$ and a stride of 10. Similarly to the WikiText case studied in the main text, we observe semantic clustering of word embeddings and that circular structures appear both when taking the PCA of the data and the trained encoder weights.

# F    WEIGHT DECAY AND SUPERPOSITION

The main text argues that weight decay favors solutions that exploit shared low-rank structure because these achieve good reconstruction with smaller weight norm than feature-by-feature interference filtering. The extended UMAP comparison in Figure 9 supports this interpretation: semantic structure remains visible across a broader range of latent sizes when models are trained with weight decay.

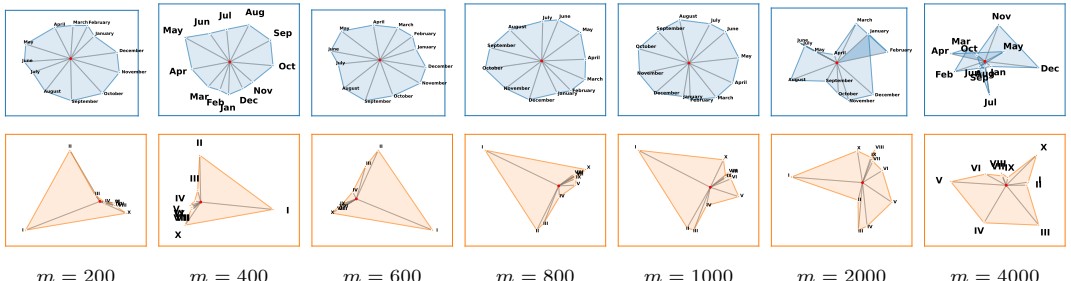

| $m = 200$ | $m = 400$ | $m = 600$ | $m = 800$ | $m = 1000$ | $m = 2000$ | $m = 4000$ |
|---|---|---|---|---|---|---|

Figure 13: Reconstructions at different latent-vector sizes. Top: "Months" dataset; middle: "Roman numerals"; bottom: corresponding latent size $m$.

## G MORE DETAILED EXAMPLE OF GROUPS OF FEATURE GEOMETRY IN BOWS

In the main paper we only show the feature structures for some representative latent sizes due to space constraints. In Figure 13 we show the structures studied in Figure 6 for an extended range of latent sizes.

We also show a zoomed in version of one of the UMAP plots in Figure 16. This figure highlights the rich structure of the features beyond simple clustering of high-level classes. We see that words corresponding to sciences are clustered together, but within this high level cluster, sub-groups like words about medicine (top left), astronomy (lower left), chemistry (lower center) and biology (center) are also grouped in smaller clusters.

### G.1 SOME EXAMPLES BEYOND 2D

Beyond the 2D examples presented in the main paper, we include 2 examples showing that the days of the week and months of the year have structure beyond a 2D circle (Figure 14). This is clear in the case of the months where an ondulation in the third principal component is present beyond the 2D circular structure.

## H OTHER CORRELATION STRUCTURES IN SYNTHETIC DATA

In Figure 15 we show the superposition patterns for the values of $m$ missing in Figure 1, as well as examples for autoencoders trained on data with a figure-of-eight correlation structure or a spherical structure. In all 3 cases we see that the Gramm matrix structure is similar in the linear and ReLU cases for $d = 2$ and $d = 3$ but they diverge at larger latent sizes as the ReLU-AEs start leveraging non-linear superposition which is indicated by sparse interference patterns (for example for $d = 8$).

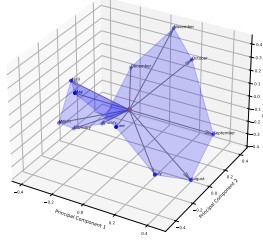

(a) Months

We reuse the "synthetic months" pipeline described in Appendix A but change only the latent curve $z(\cdot)$ and the feature directions $W$; Steps 1 (phase selection), 4 (Bernoulli sampling), and the log-odds $\ell_k = \beta W_k^\top z + b$ are identical, with defaults $\beta = 5.0$, $b = -2.0$, noise $= 0.1$, seed $= 42$.

For the *Figure-8 (Lissajous)* (Lawrence, 1972), we replace Steps 2–3 by:

$$z(\theta) = \begin{bmatrix} \sin\theta \\ \sin(2\theta) \end{bmatrix}, \qquad W = \Big[(\sin\varphi_k, \sin(2\varphi_k))\Big]_{k=0}^{F-1}, \quad \varphi_k = \tfrac{2\pi k}{F}.$$

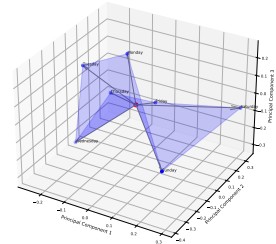

(b) Days

For the *Sphere* ($\mathbb{S}^2$), we replace Steps 2–3 by a 3D unit-sphere embedding:

$$z \sim \text{Unif}(\mathbb{S}^2), \quad W = \Big[(\cos\theta_k \sin\phi_k, \ \sin\theta_k \sin\phi_k, \ \cos\phi_k)\Big]_{k=0}^{F-1},$$

where a Fibonacci lattice (Marques et al., 2013) gives approximately uniform feature directions:

$$\phi_k = \arccos\Big(1 - \tfrac{2(k+0.5)}{F}\Big),$$

$$\theta_k = \pi(1 + \sqrt{5})\,(k + 0.5).$$

Figure 14: 3-D PCA of the embeddings for the words and the days in a WikiText BOWS setup.

Full implementation details are provided in the supplementary material.

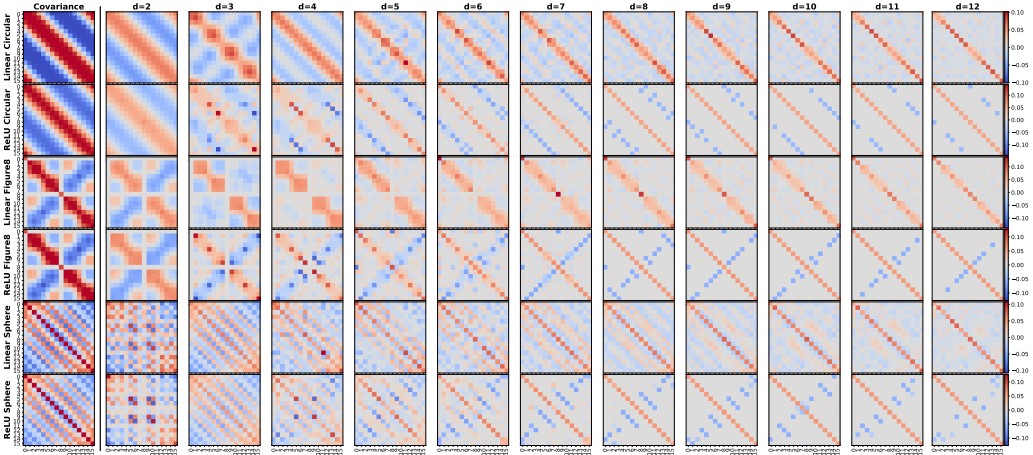

Figure 15: Extension of Figure 1 to include all values of m between 2 and 12, as well as a comparison with the weight patterns for AEs trained on data drawn from data with different correlation strucutres: figure-of-eight and spherical.

## I   A TAIL OF PARTIALLY RECONSTRUCTED FEATURES

An important implication of our results is that meaningful geometry does not necessarily imply that a feature is accurately represented. In fact, the opposite can occur: when capacity is limited, a model may place a feature in the correct semantic region because it is projecting it onto shared directions used by correlated features, while still failing to capture most of its individual variance. In this sense, semantically meaningful geometry can reflect a best guess induced by surrounding features rather than a precise representation of the concept itself.

This is what we observe in Figure 17. Even when we restrict to features with poor reconstruction scores ($R^2 < 0.3$), they still form semantic clusters. A natural explanation is that the autoencoder has learned a low-rank representation of dominant co-activation structure and projects many features, including relatively uncommon ones, onto this shared subspace. Such features inherit meaningful geometry from the data statistics, even when the model does not allocate enough capacity to reconstruct them accurately.

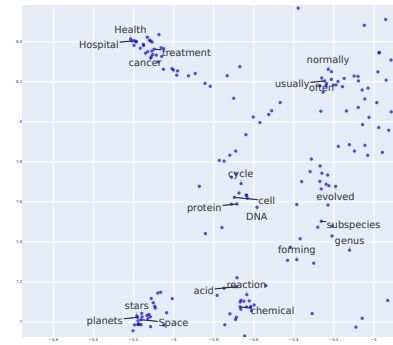

Figure 16 provides a concrete example of this effect. The figure shows a zoomed-in portion of the UMAP for the model with latent size $m = 200$, a regime in which many words in these clusters are still only partially reconstructed. Nevertheless, the local geometry is highly meaningful: words related to medicine, astronomy, chemistry, and biology form distinct sub-clusters within the broader science region. This suggests that the model has already learned where these features belong relative to one another before it has sufficient capacity to represent each of them accurately on its own. In other words, shared structure in the data can organize the representation geometry well before it supports accurate feature-level recovery.

Figure 16: Zooming into the cluster for science features in the UMAP plot with a latent size of 200, we observe sub-clusters within it. Medical features are in the top left while astronomy features are in the lower left and chemistry features are in the lower center.

## J   IMPLICATIONS FOR THE LINEAR REPRESENTATION HYPOTHESIS

While the linear representation hypothesis (LRH) is one of the pillars of current mechanistic interpretability (MI) approaches. There is still no consensus on the correct formulation of this hypothesis. The LRH can be taken to mean that internal features of a model correspond to activations along one-dimensional directions in activation space (Engels et al., 2025). However, the LRH can also be formalized around the mathematical notion of linearity meaning the representation of two features is the addition of their representations and scaling a feature corresponds to scaling its representation (Elhage et al., 2021).

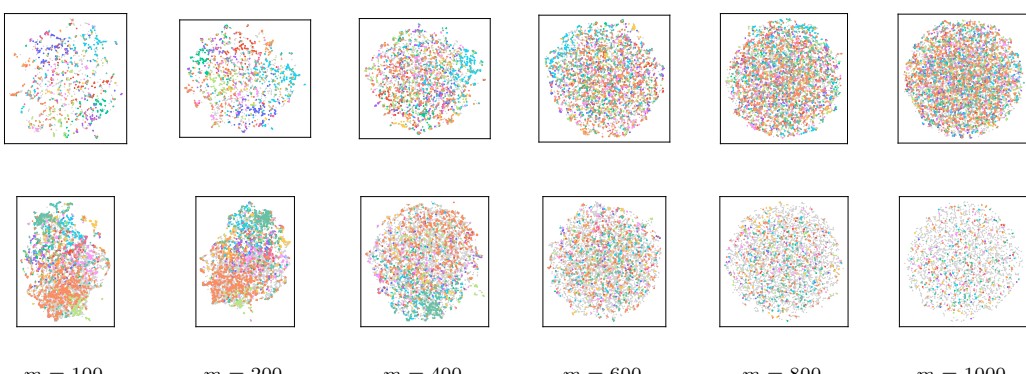

$m = 100 \qquad m = 200 \qquad m = 400 \qquad m = 600 \qquad m = 800 \qquad m = 1000$

Figure 17: UMAP embeddings at different latent-vector sizes including only features with $R^2 < 0.3$ (top) and $R^2 > 0.3$ (bottom). Semantic clusters at different latent sizes are still observed in both, although this effect is combined with an increase in the number of features above the threshold in the lower one.

While some works have suggested that observed feature geometry like the ordered circles formed by the months undermine the first definition (Engels et al., 2025; Sharkey et al., 2025), our results show that these structures can emerge from the compression and reconstruction of one-dimensionally linear features. This means that these structures do not necessarily undermine either formulation of the LRH. On the other hand, our results in Section 5 do suggest that some features used by DL models can be value-coding meaning they can encode concrete trigonometric values or coordinates along linear directions which do not fulfill the constraints for mathematical linearity. For example scaling the value of a cosine-coding feature leads to a different (and potentially invalid) cosine value, rather than a stronger activation of the same cosine value.

An interesting line of research would be to explore if presence-coding features can have value-coding components. Findings like the fact that city representations in language models can be projected linearly onto a coordinates subspace (Gurnee & Tegmark, 2024), or that integers can be projected onto a helix subspace (Li et al., 2025) could be understood through this lens. In this view, city representations could have a coordinate-coding component and integers could have a size-coding component as well as sine and cosine coding components which combine to make a helix structure.

Overall, our findings show that rich feature geometry can be explained away by linear superposition recovering the structure inherent in the data, without appealing to non-linearly encoded information with a functional role in calculation. However, we believe the existence of value-coding features could be in conflict or an exception to features being mathematically linear.

## K  OTHER DIMENSIONALITY REDUCTION METHODS

To verify that the semantic clusters are not dependent on the choice of dimensionality reduction method, we include t-SNE (van der Maaten & Hinton, 2008) and PaCMAP (Tuncer et al., 2015) as two alternatives in fig. 18.

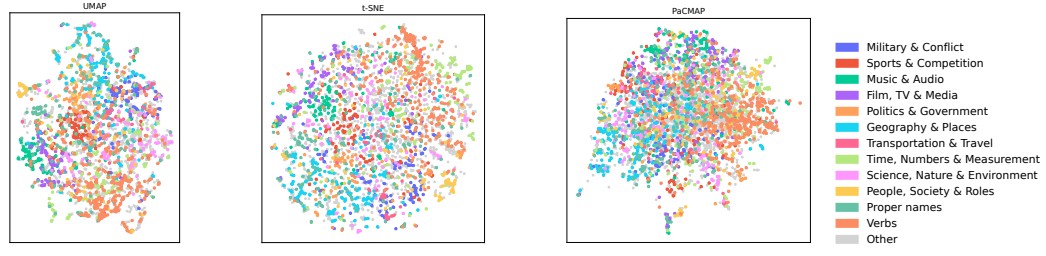

Figure 18: We show the latent representations of the top 4000 most frequent words using UMAP (left) t-SNE (middle) and PaCMAP (right) to highlight that these semantic clustering results are not dependent on the choice of dimensionality reduction technique.

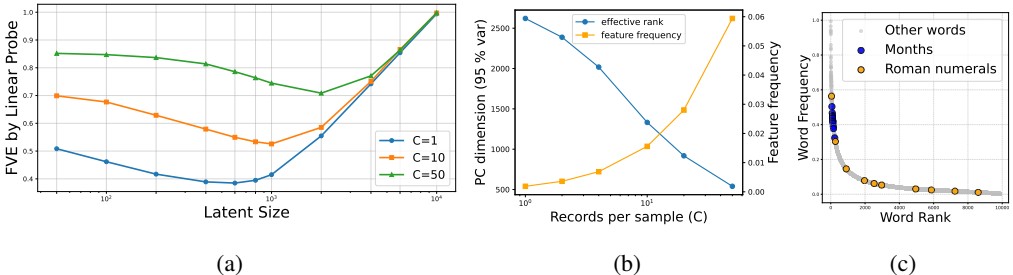

(a)  (b)  (c)

Figure 19: **Larger context windows strengthen shared structure and make constructive interference more effective. (Left)** Increasing the amount of text encoded by each sample ($C$) increases the fraction of ReLU-AE reconstruction variance that can be recovered by a linear probe from the latent space. **(Middle)** Increasing $C$ decreases the number of principal components needed to explain 95% of the variance while increasing the average number of active features per sample. **(Right)** Word frequencies follow a power-law distribution (Zipf, 1949), helping explain why some feature groups, such as months, receive more capacity and retain structured geometry longer than rarer groups such as Roman numerals.

## L  CONTEXT SIZE, EFFECTIVE RANK, AND FEATURE FREQUENCIES

The main text argues that constructive interference becomes more useful when the data contains stronger shared structure. One simple way to increase such structure in BOWS is to enlarge the context window used to construct each sample. This makes co-occurrence patterns denser and more informative, increasing the extent to which reconstruction can be supported by shared low-rank directions rather than only by feature-specific interference filtering. Figure 19 quantifies this trend from three complementary perspectives: the linearity of the learned reconstructions, the effective rank and sparsity of the data, and the frequency heterogeneity of words that helps explain why some groups of features retain structured geometry longer than others.

