# OpenReview forum: "From Data Statistics to Feature Geometry: How Correlations Shape Superposition"
_ICLR.cc/2026/Conference — ICLR 2026 Poster_

### Official Review · Reviewer_vBLf · 2025-10-18

**Soundness:** 2
**Presentation:** 2
**Contribution:** 2
**Rating:** 2
**Confidence:** 3

**Summary:**

This paper introduces BOWS, which induces the formation of complex structures of features in an autoencoder network that tries to compress and reconstruct binary bag-of-words vectors. With both a linear and nonlinear autoencoder, BOWS shows that networks do not need any goals besides compression alone in order to form these structures. The feature structures form as a result of feature correlations in the data, which are exploited by the autoencoders for compressive purposes. Experiments show the successful recovery of feature structures known to exist in LLMs (such as circular month-of-the-year features and latitude-longitude features), from the learned autoencoder representations. The authors offer a distinction between value-coding and presence-coding features, and argue that value-coding features are used to perform computation. BOWS sheds some light on the reason that LLMs form complex feature structures through superposition, arguing that feature compression, without any additional goal, is the cause.

**Strengths:**

- The paper provides convincing evidence that complex structures of features can emerge purely from the task of compressing features for storage, when there are correlations in the feature distribution.
- Figure 6 provides convincing evidence that complex structures of features form and then disappear as the latent size changes, when training a network to compress information.

**Weaknesses:**

The paper is unnecessarily complex and convoluted for it's goal in helping interpretability efforts.
- BOWS is much less a "framework" than an application of a pre-existing technique for demonstrating superposition, in Elhage (2022) [1], but on a custom bag-of-words dataset. Most of the "framework" parts of the work (techniques for analysis of results, PCA, visualization, training methodology) are inherited from Elhage (2022).
- The framework's goal is to show that compression alone can be responsible for complex feature structures learned in LLMs, but this can already be shown by taking a PCA of the data (implicit compression), making the autoencoder analysis an unnecessarily complicated solution.
- The distinction made between presence-coding and value-coding neurons is weak/useless. The difference as defined seems to boil down to an artifact of the way the English language treats some things as quantifiable/measurable and others as not, rather than intrinsic to an LLM's internal functioning, so it is trivially true that there exist neurons for both presence-coding and value-coding things. The "value of the north-south coordinate" is just "the presence of northerliness". While this may seem contrived, there are plenty of fuzzy edge cases to be made ("lots of hair" (presence) vs "many hairs" (value)).
  - Altogether, I do not see why making this distinction of quantifiableness helps us understand the internal functioning of LLMs any better than before. In general, I don't see how the mathematical behavior of the autoencoder is supposed to depend on things outside of the correlation structure of the features, such as our external understanding of the feature's meaning as the presence vs value of something.
  - If the authors want to instead define presence-coding versus value-coding based on how the features are being used, they should clarify precisely what conditions are sufficient to qualify the feature as one or the other. An example does not suffice as a definition, as in lines 343-347.
  - The authors give no examples of how making this distinction helps us to reason about other aspects of the features. Being able to identify the features as "value-coding" does not help tell us how it's used. Rather, identifying how it's used allows us to label it as "value-coding", and there doesn't seem to be any purpose served after that.

[1] Elhage, et al., "Toy Models of Superposition", Transformer Circuits Thread, 2022.

**Questions:**

I am presuming below that as indicated in the abstract, the main goal of the paper is to show that compression alone can be responsible for complex feature structures in LLMs, when features statistics are correlated.
- It seems like this goal is already achieved with just PCA, so why ever bother with the autoencoder?. In what case does the autoencoder reveal any structures to you that PCA/probing does not? As you mention in Section 3.1, the linear autoencoder will be recovering the top two principal components anyways.
- It seems like you designed the autoencoder as the simplest possible example where compression can be demonstrated as the root cause for the emergence of the feature structures. Why complicate the matter with a nonlinear autoencoder? Is it to take into account the possibility that the presence of nonlinearity can disrupt the usage of the complex structures as the most efficient compressed representation?

Much of the rest of the analysis describes how the autoencoders behave, but this is not necessary in order to see that the structures are inherited from data statistics, since it is already evident from PCA and probing.

For line 397, is there converse evidence that the presence-coding features are _not_ used? Also, I find this experiment unconvincing of the importance of the difference between value-coding and presence-coding. It looks like you are defining the features that are known to be used in a particular way as the value-coding ones, and the rest as presence-coding, and then using that to argue that the value-coding ones are the useful ones, which I do not see the point of.

Random:
- Figure 7 right: You are trying to probe one city's coordinates from an 8 dimensional vector which predicts the relative positioning between two cities. Which of the two city's coordinates are you trying to probe, and which city is plotted?
- Figure 3 typo: (a-c) instead of (a)
- Please try to keep the figures placed close to where they are referenced.

---

> ### Author Response · Authors · 2025-11-27
> **Author response**
>
> We thank the reviewer for the constructive feedback below we clarify the goal of the paper and address the reviewer's questions and concerns.
>
>
> **The goal is not 'already achieved with just PCA'** Our goals are to 1) present a _new way to think about superposition that includes linear superposition_ 2) show that _linear superposition can emerge in non-linear models such as ReLU-AEs_ under realistic circumstances (correlations and weight decay) 3) show that _linear superposition explains previously observed structures_ in the activation space of LLMs. While a simple PCA plot could have shown that this structure was implicit in the data covariance, the key contribution of the paper is to study under what circumstances non-linear models leverage linear superposition, giving rise to this kind of structures. Based on reviewer feedback we realize that 1) should have been made more rigorous and 3) would benefit from experiments beyond simple autoencoders. We therefore made the following changes to the manuscript.
>
> To reinforce 1) and make this clearer, we now formalize our definition of superposition such that it includes linear and non-linear superposition as special cases (Section 2.1). We then provide mathematical analysis for how linear and non-linear superposition handle interference and why weight decay combined with tight bottlenecks introduces a bias for linear superposition even in non-linear models (Section 3.1). While previous work sees interference arising from superposition as requiring non-linear filtering, we show that this need not be the case when the data covariance has low rank or fast spectral decay as in realistic data. In these cases, we show that capturing the principal components of the data corresponds to arranging features such that interference aligns with the signal and contributes to an accurate reconstruction.
>
> To reinforce 3) and highlight the implications of our work for the kind of superposition we can expect in real models, we replicated the main results in a 2-layer transformer model with word level tokenization trained with BCE loss to recover the set of distinct words present in the context up to that word. Appendix B now shows that the same circular structures and semantic clustering appear in the residual stream of this transformer models further establishing that linear superposition appears in non-linear models.
>
>
> **Lack of novelty** While our BOWS is architecturally very similar to the toy models in Elhage et al. 2022, we highlight that by introducing realistic elements like leveraging the correlations in internet text and studying the effect of weight decay, we observe completely different superposition patterns. Where Elhage et al. (2022) observe local regular polytope structures, we observe global semantically meaningful structure. The fact that this happens in a non-linear model is key in both our work and Elhage et al. since the goal is to understand how information is represented in real deep learning models which are non-linear. We hope this is made clearer by our new experiments on transformers which highlight linear superposition in the residual stream.
>
> **Is BOWS a framework?** We believe BOWS is an interesting setting to study superposition. It can serve as a framework in that different datasets, window sizes, vocabulary sizes and weight decay values lead to different superposition structures. While the autoencoder setting is the same, we believe the additional variables we introduce make it a distinct framework rather than an instance of the toy models setup.

---

> ### Author Response · Authors · 2025-11-27
> **Author response (part 2)**
>
> **Value coding features:** The goal of Section 5 is to show that while correlations in the data are sufficient to create semantically meaningful structure in the representations, they are not necessary. To show this we highlight that a map of the US appears in the model's representations when the only task is to know the direction (e.g. North East) from one US city to another. This happens because the model learns to represent the $x$ and $y$ coordinates of the cities in order to compare them and determine the direction from one to the other (note that Section 5 does not use the autoencoder setting). We have now provided clearer definitions for presence-coding and value-coding features which we do believe to be a useful distinction. For example a 'lots of hair' feature could activate on images of people with lots of hair without having a linear representation that can be used to read off the exact number of hairs that person has. Similarly, our model could have learned a presence coding representation of 'east coast" and 'west coast' from which we wouldn't be able to extract precisely how far east a city is. While any value-coding feature can be turned into a presence-coding feature where the value is above a threshold, not every presence-coding feature is a value-coding feature. The reason we are interested in value-coding features is that by linearly representing meaningful values (like coordinates), they give rise to complex structure like a map, whereas more coarse presence-coding features may only have given rise to clusters around east, west, north and south. We have updated the paper to make this distinction clearer with more formal definitions for these 2 concepts:
>
> **Presence-coding features:**
> >We say that a representation $h(x) \in \mathbb{R}^d$  contains a _presence-coding feature_ if some binary or categorical variable
> $y(x)$ (e.g.\ ``this token is the word \textit{cat}'') is recoverable by a linear classifier. Formally, there exist weights
> $\{w_k, b_k\}$ such that $\hat y(x) = \arg\max_k (w_k^\top h(x) + b_k)$ predicts $y(x)$ with low error. Presence-coding features
> thus behave as detectors for discrete properties, and different values of $y$ are treated as separate classes without requiring
> any particular geometric relation between them in representation space \emph{a priori}. For presence-coding features structured representations are contingent on correlations in the data and capacity constraints that lead the features to be represented in linear superposition.
>
>
> **Value-coding features:**
> >In contrast, we say that a representation $h(x)$ contains a _value-coding feature_ if a real-valued variable
> $v(x) \in \mathbb{R}$ (e.g.\ an angle, a coordinate, or a continuous latent factor) is linearly decodable. That is, there exist
> $w \in \mathbb{R}^d$ and $b \in \mathbb{R}$ such that $\hat v(x) = w^\top h(x) + b$ approximates $v(x)$ with low error. A collection of such value-coding features $v_1(x),\dots,v_k(x)$ defines a low-dimensional \emph{value space} $\mathbb{R}^k$; plotting examples in this space can reveal semantically meaningful structures (for instance, a 2D map from $(x,y)$ coordinates, or a circle from $(\sin \theta, \cos \theta)$ in modular addition). Crucially, these structures are fully accounted for by the existence of linear value codes for the underlying variables and therefore exist even in the absence of superposition.

---

### Official Review · Reviewer_2aH1 · 2025-10-26

**Soundness:** 2
**Presentation:** 1
**Contribution:** 2
**Rating:** 2
**Confidence:** 3

**Summary:**

This work studies correlations between (input) features in a “toy models of superposition” (Elhage et al. 2022) type setting, and introduces a distinction between linear and non-linear superposition.  While Elhage et al. focused on non-linear superposition, this work argues that linear superposition is more common in real data.

The submission also introduces a Bag-of-Words Superposition (BOWS) setup, which uses co-occurrence in real documents to generate correlated binary feature vectors.  (BOWS is a bit of a misnomer since bag-of-words representations capture multiplicity, but the representation in this work does not).  The work also postulates that these co-occurrence statistics may be largely responsible for the particular geometric patterns observed in LLM representations (e.g. months being arranged in a circle), although I don’t believe this hypothesis was actually tested.

**Strengths:**

The observation that linear superposition is more common (if correct), seems important.

The BOWS setup seems like a nice, sensible approach to introducing realistic correlation structure, and has the bonus of enabling researchers to bring their knowledge of words’ semantics to the analysis of experiments.

**Weaknesses:**

(major): I believe this work is in need of more rigor in establishing its central definitions and concepts (e.g. “superposition”, “linear superposition”).  For instance, I’m not sure what is the content of the statement “This explicitly shows that linear dimensionality reduction enables a form of superposition (d = 12 > m = 2) by exploiting feature correlations, without requiring any non-linearity.”  This just sounds like it’s saying “you can reconstruct an input well using a few principle components when input dimensions are highly correlated”, but that’s nothing new.  Similarly, “features in linear superposition inherit their structure from their covariance matrix” seems vacuous if linear superposition basically just means PCA.  While superposition is a more established concept, it relies (in my experience) on assuming some ground-truth set of underlying features.  In this case, I guess these are the words in the BOWS representation, but this should still be made more explicit.  I would like the discussion of superposition and linear superposition to be grounded in a rigorous mathematical exposition of the concepts.

(major): The submission claims that Elhage et al. (2022) don't study linear superposition or correlated features. However, Elhage et al. do consider correlated features, and state “when there isn't enough space to represent all the correlated features, it will collapse them and represent their principal component instead”.  This is a significant mischaracterization of key related work, and makes me question the novelty of this submission.

(moderate): I think some of the statements in this submission overstate the successes of mechanistic interpretability, eg:
- “These approaches have successfully uncovered interpretable units corresponding to semantic concepts, syntactic roles, or specific input patterns.”
- “Initial works in MI studied interpretable monosematic neurons”
I think referring to neurons as 'interpretable' is generally accurate.  They may seem more or less interpretable, but so far I’ve not seen sufficient evidence to justify claims that we really understand what neurons are doing, except perhaps in the context of particular tasks / distributions of inputs.  This is an important distinction, as overstating the success of interpretability can provide false assurances.

(minor): The related work section should do more to connect the referenced works to the submission.


(nit): The acronyms SDL and LRH should be introduced.

**Questions:**

“However, polytope-like structures have not been observed in standard LLM activations” seems to refer specifically to Elhage et al.’s 2022 work; however, Park et al. (2025) claim to have found polytopes in the activations of Gemma.  Does this invalidate the quoted claim?  How are the polytopes discussed in Park et al. related to those from Elhage et al. (2022)?

How do the experiments in Section 5 differ from what was already done by Gurnee & Tegmark (2024)?

“In contrast, the patterns in Figure 7 emerge despite inputs being uncorrelated,” Can you please make the claim about inputs being uncorrelated precise and support it?

I’m not convinced that “circles and clusters resemble the global semantic organization first reported in distributional word embeddings”.  Can you elaborate on and substantiate this claim?

The abstract states: “Our findings suggest that the semantically meaningful structures observed in language models could arise driven by compression alone, without necessarily having a functional role beyond efficiently arranging feature representations.”  What would it mean for this to be the case?  Are there experiments that could falsify this as a hypothesis?

---

> ### Author Response · Authors · 2025-11-27
> **Author response**
>
> We thank the reviewer for taking the time to review our work and for their constructive feedback. Below we adress the reivewers questions and concerns:
>
> **Mathematical rigor and clear definitions:** We thank the reviewer for highlighting this. We have now introduced a new Definitions subsection (Section 2.1) to introduce superposition (linear and non-linear) and the linear representation hypothesis. We have also provided mathematical analysis of how interference is handled under linear and non-linear superpositions, to provide valuable intuitions regarding the well-studied behavior of linear autoencoders in terms of superposition and interference. We have made it clearer that our contribution adds to existing work by 1) Framing PCA as a kind of superposition and giving an intuition for how interference is handled by PCA, but mainly 2)_Studying under which circumstances non-linear models leverage linear superposition despite their non-linearities_. While 1) is a potentially useful framing of PCA in terms of superposition, the main contribution is 2) as it speaks to why we should expect this kind of superposition in non-linear models. To make this point clearer and go beyond our BOWS setting we have added a transformer-based word recovery task in which we observe the same dynamic of linear superposition emerging in the presence of weight decay leading to circular configurations for the months of the year and semantic clustering of features (Appendix B).
>
>
> **Similarity to Elhage et al. (2022):** We thank the reviewer for highlighting the lack of clarity on this point. We note that Elhage et al. see superposition as introducing interference that needs to be non-linearly filtered out. As such, they see PCA as in conflict with superposition since all the principal components of the covariance need to be captured in their setting in order to achieve an accurate reconstruction. However, it is true that they do introduce pairwise correlations and have some relevant insights on their impact on superposition. We have now updated the paper to properly introduce this in the background section and in a footnote in section 2.1.
>
> >Elhage et al. 2022 also consider pairwise correlated features, but in their setup, all principal components carry significant variance and PCA collapses correlated pairs onto indistinguishable points. However, when $\Sigma$ is approximately low-rank, fewer principal components suffice making linear superposition viable.
>
>
> **Overstating success of mechanistic interpretability:** We agree with the reviewer that neurons are not generally interpretable. We refer to works claiming this as 'Initial works' to contrast with later advances such as superposition and SAEs, that suggest that most neurons are polysemantic, and hence not interpretable. This has lead the field to look for interpretable directions in activation space rather than neurons. We have made this point clearer in the related work. Similarly we have qualified the statement about ``interpretable units'' by clarifying that the units are directions in activation space, not neurons, and that this is only true of some concepts.
>
> **Acronyms:** We thank the reviewer for pointing this out, we have now updated the paper to properly introduce these acronyms.
>
> **Questions:** In line 53 we missed the key qualifier that *regular* polytopes have not been observed in LLM activations. This is crucial since any set of vectors can be seen as a polytope, while the regularity is a specific claim about features arranged to minimize pairwise interference. Park et al. (2025) project animals onto the representations for different animal categories like `mammal' forming non-regular polytopes. Park et al. (2025) Figure 9 shows that these categories all have positive cosine similarities with each other and are therefore not arranged in local regular polytopes. We have now fixed this missing qualifier in line 53.
>
>
> **Clarifying 'without having a functional role beyond efficiently arranging feature representations':** In this work we show that efficient compression of feature representations is *sufficient* to give rise to these structures. We believe this is an important null hypothesis to explain feature geometry when the semantically meaningful structure aligns with the data covariance. However, this does not mean that feature geometry never has a functional role. For example, our experiments in Section 5 aim to show that correlations in the data are a sufficient but not necessary condition for semantically meaningful structure in the representations as explained in our following answer.

---

> > ### Author Response · Authors · 2025-11-27
> > **Author response (part 2)**
> >
> > **Difference with Gurnee and Tegmark (2024) and uncorrelated inputs:** While the observations of Gurnee and Tegmark (2024) inspired our map setup for Section 5, our contribution is to isolate a sufficient cause for this structure to emerge. As opposed to structures like the circles formed by the months of the year which we show can be explained by their correlations, we show that this map-like structure emerges naturally for the functional role of predicting the relative positions of  cities. In our setting the input is a pair of cities and the output is the relative cardinal direction from one to the other. Therefore, any pair of cities appears only once either in the training or the validation dataset so no 2 cities are correlated in the input. We have now made this clearer in the main text.
> >
> > **Connection to distributional word embeddings:** We aim to highlight that distributional embeddings also exhibit semantic clusters, in that words with similar meanings are embedded closer to each other. As we note in the related work, these kinds of embedding methods can also be connected to PCA. However, it is true that to our knowledge there are no studies of circular structures in these kind of embeddings. To avoid confusion on this point we now word the sentence as follows:
> > >'Intriguingly, such semantic clusters resemble the global semantic organization first in distributional word embeddings'

---

### Official Review · Reviewer_WkYu · 2025-10-29

**Soundness:** 4
**Presentation:** 4
**Contribution:** 3
**Rating:** 8
**Confidence:** 3

**Summary:**

The paper investigates the geometric structure of internal features in mech interp, aiming to bridge the gap between "toy model" theories of superposition and observations in real-world language models. Prior work suggested superposition creates interference-managing structures (like polytopes), but empirical studies of LLMs instead find semantically-rich structures (like semantic clusters or ordered circles for months).

The authors introduce a framework called Bag-of-Words Superposition (BOWS), where a simple ReLU autoencoder is trained to compress high-dimensional, sparse, binary bag-of-words vectors derived from real text. The use of the BOW space in such analysis is interesting as it provides a proxy for the platonic, objective, or "ideal" embedding space for features that we cannot access.

The paper makes two key claims:

It distinguishes between non-linear superposition (as seen in toy models), which uses non-linearities like ReLU to create local structures (e.g., antipodal pairs) to manage interference between uncorrelated features, and linear superposition, which emerges when features are correlated.

In the linear superposition regime—which is induced by tight bottlenecks or, significantly, by weight decay—the non-linear AE simply learns to linearly encode the low-rank structure (i.e., the principal components) of the data's covariance matrix.

The authors demonstrate that this linear superposition mechanism is sufficient to reproduce the exact semantic clusters and circular representations seen in LLMs. This suggests these structures are a "parsimonious" explanation, arising merely as a byproduct of efficient compression, and may not have a specific functional, computational role.

**Strengths:**

1, The paper's primary strength is its clarity. The distinction between "linear superposition" (PCA on correlated data) and "non-linear superposition" (local, ReLU-dependent structures for uncorrelated data) is a very useful and clear conceptual framework. And the flow of the paper is natural, too.

2, The BOWS framework offers a good trade-off, it's more realistic than toy models but far more controllable than a full LLM. Using it to show the emergence of semantic clusters and circles from data statistics alone is highly effective. The experiment showing that a ReLU AE transitions from a linear (PCA) regime to a non-linear (antipodal) one as the latent dimension $m$ increases is clean and convincing.

3, This work provides an interesting for feature geometry. It challenges the assumption that structures like semantic clusters or "month circles" are necessarily functional or computationally constructed by the model. The idea that they are simply a byproduct of compression (driven by data correlations and weight decay) is a fundamental claim that the mech interp community must consider.

**Weaknesses:**

1, The main limitation is the simplicity of the BOWS setup. An AE trained on static BoW vectors does not necessarily capture all the SAE variants that exist in literature today, since many of them explicitly make architectural changes to cater to the space where features live.

2, Also, the paper seems to strongly implies that these structures are just byproducts and not functional. This dichotomy might be false. A model could (and likely would) exploit this emergent, PCA-driven structure for computation. Framing it as byproduct OR functional is perhaps too strong and I think the two are not mutually exclusive.

**Questions:**

My question would be whether "byproduct" structures also be "functional"? Couldn't the model leverage the fact that the optimal compression scheme (PCA) naturally arranges features in a computationally convenient way (e.g., a circle for months)? Or perhaps a little bit more challenging (in future works), what happens when the low-rank structure does not facilitate (or even adverserial for) a geometry that makes underlying computation (like modular arithmetic) easier?

My question, however, does not impact my perception of this paper as a novel and solid contribution to mech interp community.

---

> ### Author Response · Authors · 2025-11-27
> **Author response**
>
> We thank the reviewer for the positive and constructive feedback. Below, we address the reviewer's questions and concerns:
>
> **Simplicity of the BOWS setup:** Our intention with BOWS is to model how transformers read and write information to their residual stream, and more generally to study the structure of features encoded in superposition under realistic conditions like correlations in the data and weight decay. To make this clearer and go beyond the simplicity of the BOWS setup we have now replicated the main results of the paper in a simple transformer setting which takes in text tokenized at a word level and is trained to reproduce the set of words that are present in the input from a vocabulary using binary cross-entropy loss (multi-class). Similar to BOWS, this is a pure reconstruction or recall task in a non-linear model, yet we observe linear superposition in the residual stream of this transformer, as evidenced by the circular arrangements of the month representation or the broader semantic clustering of words. The connections to different SAE architectures is interesting: we think BOWS could be used as a setting for evaluation and comparison of different SAEs.
>
> **'Functional' vs 'byproduct' dichotomy:** We agree with the reviewer that the tradeoff between the efficiency of representations and their functional role is an important avenue for future work. We do expect that there could be cases where 'byproduct' structures are also 'functional'. For example it seems likely that some co-occurring words have similar continuations and therefore similar functional roles. We have added the following passage to the discussion to bring this to the attention of the readers.
>
> >'Additionally, while we show that structured representations do not require a functional role beyond efficient compression, this does not need to be a dichotomy as this kind of structure could be both functional and efficient.'

---

### Author Response · Authors · 2025-11-27
**General response**

We thank the reviewers for their time and the constructive feedback on our work. We believe addressing these reviews has enabled a substantial improvement of the manuscript. Below we highlight the main updates to the paper, and address each reviewer's concerns in their individual responses.


**Definitions and mathematical grounding.** We have used the additional page provided for the rebuttal to include formal definitions for concepts like linear and non-linear superposition. We provide a definition of superposition that is not restricted to sparse iid. data and can encompass linear and non-linear superposition as special cases (Section 2.1). As encouraged by reviewer [2aH1], we also provide a more mathematically grounded discussion of linear superposition making a clear connection to PCA and providing a clear intuition for why correlations in the data allow models to harness interference rather than non-linearly filtering it out.

**Beyond the toy setting.** To explicitly demonstrate the implications of our results for real models, we now show that this same behavior we studied in BOWS is replicated in a transformer with word-level tokenization trained to reproduce the set of unique words in the context up to the current token with BCE loss (full details in Appendix B). In this setting, we also observe circular arrangements of the features corresponding to the months of the year, as well as semantic clustering in the residual stream as a byproduct of compressing the vocabulary of 10,000 words into a residual stream of 768 dimensions when training with weight decay.

**Connection to Toy Models of Superposition (2022).**  While Elhage et al. [2022] briefly discuss a setting with 3 pairs of correlated features embedded in a 2d space, they do not explore data with more complex and realistic correlations and leave the exploration of the relationship between superposition and PCA for future work. Elhage et al. see superposition as requiring non-linear filtering, since the kind of correlations they introduce do not lead to the kind of low rank structure that can be captured by linear superposition or PCA. We now make this clearer when introducing the BOWS setup (Section 2.2) and provide the following footnote in Section 2.1:

>Elhage et al. 2022 also consider pairwise correlated features, but in their setup, all principal components carry significant variance and PCA collapses correlated pairs onto indistinguishable points. However, when $\Sigma$ is approximately low-rank, fewer principal components suffice making linear superposition viable.

---

### Author Response · Authors · 2025-12-03
**Rebuttal summary**

Dear AC,

We would like to take this opportunity to provide an overview of the rebuttal to inform your final decision. While reviewers did not get a chance to respond to our rebuttal, we believe we addressed their main concerns and clarified aspects of the paper that had caused some confusion. In particular:

-	In response to reviewer [2aH1] we provided more formal definitions for key concepts in our work such as linear and non-linear superposition (Section 2.1). We also provided a more mathematically grounded discussion of linear superposition which provides a clear intuition for why interference does not need to be filtered out in this case and why weight decay favours linear superposition even in non-linear models (Section 3.1).

-	Our initial submission did not make the relevance of studying linear superposition in *non-linear* models and its implications for the kind of superposition we can expect in real deep learning models clear enough. This led to concerns from reviewers [vBLf] and [2aH1] about the implications of our work beyond the well studied behaviour of linear autoencoders and PCA. In particular, reviewer [vBLf] suggested that “goal is already achieved with just PCA, so why ever bother with the autoencoder?.”. To address this we have made it clearer in Section 3.1 that the key goal of the paper is to study under what circumstances an autoencoder with a non-linearity will leverage linear superposition, behaving more like a linear model. We now also provide a mathematically grounded justification for why the combination of tight bottlenecks and weight decay can lead to this regime. In addition, to make the implications of this clearer, we also replicate the emergence of linear semantically meaningful structure on the residual stream of a two-layer transformer trained to reproduce the set of words present in the context. This makes it clearer that our insights have implications for non-linear models like transformers.

-	Another concern of reviewers [vBLf] and [2aH1] was the connection to “Toy Models of Superposition” Elhage et al. 2022 . While Elhage et al. [2022] briefly discuss a setting with 3 pairs of correlated features embedded in a 2d space, they do not explore data with more complex and realistic correlations and leave the exploration of the relationship between superposition and PCA for future work. Elhage et al. see superposition as requiring non-linear filtering, since the kind of correlations they introduce do not lead to the kind of low rank structure that can be captured by linear superposition or PCA. We now make this clearer when introducing the BOWS setup (Section 2.2) and provide the following footnote in Section 2.1:

>Elhage et al. 2022 also consider pairwise correlated features, but in their setup, all principal components carry significant variance and PCA collapses correlated pairs onto indistinguishable points. However, when $\Sigma$ is approximately low-rank, fewer principal components suffice making linear superposition viable.

-	There was also some confusion around value-coding features, we now provide clearer definitions in Section 5 to highlight why we believe this is a meaningful distinction. Particularly when trying to provide a comprehensive explanation of why semantically meaningful representation structure appears in deep learning models.

We believe this rebuttal period and feedback from reviewers has allowed us to substantially improve the paper and that it would be a valuable contribution to the field and of interest to ICLR attendants. We thank the reviewers and the AC for their time and consideration.

---

### Meta-Review · Area_Chair_sAUF · 2026-01-07

**Summary:**

This work studies correlations between features and introduces a distinction between linear and non-linear superposition, showing that linear superposition is more common -- a meaningful discovery but maybe not that important. I think the proposal of the BOWS is a nice (though almost too simple) tool. The paper is well written. The original paper has the problem of being highly imprecise in mathematics, as pointed out by 2aH1. But I feel this point is improved. Given the problems with this ICLR reviewing process, I can see that this paper could be rejected or improved.

**Reviewer Concerns:**

Reviewer 2aH1's concern about mathematical precision could have been greatly improved, and is likely to raise score. But 2aH1 only gives a contribution score of 2, and so 2aH1 might either raise the score to a 4 or 6, but this is difficult to decide. I also agree that while the discovery is interesting, its significance is questionable. Essentially, I believe this work is borderline. I recommend acceptance, but there is no real reason why it could not be a rejection

**Reviewer Scores:**

See above.

---

### Decision · Program_Chairs · 2026-01-26

Accept (Poster)